# CLoVE: Personalized Federated Learning through Clustering of Loss Vector Embeddings

## Abstract

We propose *CLoVE* (Clustering of Loss Vector Embeddings), a novel algorithm for Clustered Federated Learning (CFL). In CFL, clients are naturally grouped into clusters based on their data distribution. However, identifying these clusters is challenging, as client assignments are unknown. *CLoVE* utilizes client embeddings derived from model losses on client data, and leverages the insight that clients in the same cluster share similar loss values, while those in different clusters exhibit distinct loss patterns. Based on these embeddings, *CLoVE* is able to iteratively identify and separate clients from different clusters and optimize cluster-specific models through federated aggregation. Key advantages of *CLoVE* over existing CFL algorithms are (1) its simplicity, (2) its applicability to both supervised and unsupervised settings, and (3) the fact that it eliminates the need for near-optimal model initialization, which makes it more robust and better suited for real-world applications. We establish theoretical convergence bounds, showing that *CLoVE* can recover clusters accurately with high probability in a single round and converges exponentially fast to optimal models in a linear setting. Our comprehensive experiments comparing with a variety of both CFL and generic Personalized Federated Learning (PFL) algorithms on different types of datasets and an extensive array of non-IID settings demonstrate that *CLoVE* achieves highly accurate cluster recovery in just a few rounds of training, along with state-of-the-art model accuracy, across a variety of both supervised and unsupervised PFL tasks.

## 1 Introduction

Federated learning (FL) has emerged as a pivotal framework for training models on decentralized data, preserving privacy and reducing communication overhead (McMahan et al., 2017; Konečný et al., 2016; Bonawitz et al., 2019). However, a significant challenge in FL is posed by the heterogeneity of data distributions across clients (non-IID data) (Zhao et al., 2018), as this can adversely impact the accuracy and convergence of FL models. To address this issue, Personalized Federated Learning (PFL) (Tan et al., 2023; Hanzely & Richtárik, 2020) has developed as an active research area, focusing on tailoring trained models to each client's local data, thereby improving model performance.

Our research focuses on a variant of PFL known as *Clustered Federated Learning (CFL)*. In CFL, (Ghosh et al., 2020; Werner et al., 2023; Sattler et al., 2021; Mansour et al., 2020a; Chung et al., 2022; Long et al., 2023; Duan et al., 2022; Vahidian et al., 2023), clients are naturally grouped into *K true clusters* based on inherent similarities in their local data distributions. However, these cluster assignments are not known in advance. The objective of CFL is to develop distinct models tailored to each cluster, utilizing only the local data from clients within that cluster. To achieve this goal, identifying the cluster assignments for each client is crucial, as it enables the construction of accurate, cluster-specific models that capture the unique characteristics of each group.

Many existing approaches (Awasthi & Sheffet, 2012; Kumar & Kannan, 2010) that provide guarantees on cluster assignment recovery often rely on the assumption that data of different clusters are well-separated. However, this assumption is often violated in real-world datasets, as evidenced by the modest performance of k-means on MNIST clustering even in a centralized setting. Its low Adjusted Rand Index (ARI)[1] of only approximately $50\%$ (Treder-Tschechlov, 2024) can be attributed to the

---

[1] ARI is a measure of the level of agreement between two clusterings, calculated as the fraction of pairs of cluster assignments that agree between the clusterings, adjusted for chance.

high variance in the MNIST image data, resulting in many digits being closer to other clusters than to their own digit's cluster. Furthermore, in the federated setting, clustering approaches like *k-FED* (Dennis et al., 2021) can recover clustering in one shot, but they also rely on the inter-cluster separation conditions of Awasthi & Sheffet (2012). However, as we demonstrate (App. D.1.1), even *k-FED* achieves low ARI on clustering common mixed linear regression distributions (Yi et al., 2014), as the cluster centers of those distributions can also be very close, highlighting the limitations of these approaches.

An alternative way to recover cluster assignments is to leverage the separation of cluster-specific models. For instance, the Iterative Federated Clustering Algorithm (*IFCA*) (Ghosh et al., 2020) assigns clients to clusters based on the model that yields the lowest empirical loss. However, *IFCA* has a significant limitation: it requires careful initialization of cluster models, ensuring each model is sufficiently close to its optimal counterpart. Specifically, the distance between an initial model and the cluster's optimal model must be strictly less than half the minimum separation distance between any two optimal models. In practice, achieving this distance requirement can be challenging. For example, in our experiments with the MNIST dataset, *IFCA* often failed to recover accurate cluster assignments. As shown in Fig. 1, this issue typically occurs when clients from two or more clusters achieve the lowest loss on the same model, and thus get assigned to the same cluster. This, in turn, causes the algorithm to get stuck and fail to recover accurate clusters.

In this paper, we propose the Clustering of Loss Vector Embeddings (*CLoVE*) PFL algorithm, a novel approach to tackle both the challenges of constructing accurate cluster-specific models and recovering the underlying cluster assignments in a FL setting. Our solution employs an iterative process that simultaneously builds multiple personalized models and generates client embeddings, represented as vectors of losses achieved by these models on the clients' local data. Specifically, we leverage the clustering of client embeddings to refine the models, while also using the models to inform the clustering process. The underlying principle of our approach is that clients within the same cluster will exhibit similar loss patterns,

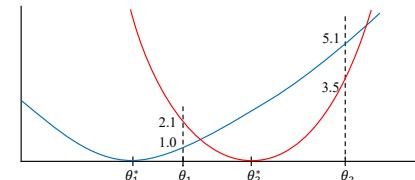

Figure 1: *Model parameters are on the x-axis and model losses on the y-axis. Blue and red curves depict losses for clients of cluster $1$ and $2$ respectively. $\theta_1^*$ and $\theta_2^*$ are optimal models for clients of clusters $1$ and $2$ respectively. Initial models are $\theta_1$ and $\theta_2$. Loss on model $\theta_1$ ($\theta_2$) is $1.0$ ($5.1$) and $2.1$ ($3.5$) for clients of clusters $1$ and $2$, respectively. Loss vectors are $[1.0, 5.1]$ for clients of cluster $1$ and $[2.1, 3.5]$ for clients of cluster $2$. In IFCA, all clients select model $\theta_1$, as it minimizes their loss. Consequently they all get assigned to cluster $1$. Moreover, because of this assignment, model $\theta_2$ receives no updates, and therefore its loss stays high; as a result, no clients ever select it in future rounds. Ultimately, this leads to all clients converging to the same cluster, failing to recover the true client clusters. In contrast, clustering based on loss vectors achieves accurate client partitioning, as the loss vectors of clients from different clusters are highly distinct.*

whereas those from different clusters will display distinct loss patterns across models. By analyzing and comparing these loss patterns through the clustering of loss vectors, our approach effectively separates clients from different clusters while grouping those from the same cluster together. This, in turn, leads to improved models for each cluster, as each model predominantly receives updates from clients within its corresponding cluster, thereby enhancing accuracy and robustness.

*CLoVE* exhibits a distinct advantage over existing CFL methods, including *IFCA*, *CFL-S* (Sattler et al., 2021), *k-FED*, *FlexCFL* (Duan et al., 2022), and *FeSEM* (Long et al., 2023). Notably, *CLoVE* achieves robust clustering results even from random initializations, a significant advantage over *IFCA*, which requires careful model initialization. Unlike *k-FED*, *CLoVE* does not rely on any assumptions on the data distribution. Compared to *CFL-S*, which often necessitates hundreds of communication rounds to reach convergence, *CLoVE* achieves clustering in fewer than ten rounds while reliably handling sparse client participation and stragglers. This accelerated convergence results in substantially lower computation and communication overhead compared to both *IFCA* and *CFL-S*. Unlike *FlexCFL*, which produces a static clustering in a single round, *CLoVE* dynamically refines its cluster assignments, offering the dual benefits of low-cost operation and the capacity to improve clustering quality over time. Moreover, *CLoVE*'s loss vector-based cluster recovery approach avoids the local optima issues inherent in expectation maximization approaches such as *FeSEM*, thereby ensuring optimal cluster recovery with high probability.

We conduct a comprehensive evaluation of our approach, demonstrating its ability to recover clusters with high accuracy while constructing per-cluster models in just a few rounds of FL. Furthermore, our analytical results show that our approach can achieve single-shot, accurate cluster recovery with high probability in the case of linear regression, thereby establishing its efficacy and robustness.

Our technical **contributions** can be summarized as follows:

- **Introduction of the *CLoVE* PFL algorithm**: We propose a novel approach that simultaneously builds multiple personalized models and generates client embeddings based on model losses, enabling effective clustering of clients with unknown cluster labels.
- **Simplicity and wide applicability**: Our method, unlike most others, extends to both supervised and unsupervised settings, handling a variety of data distributions, sparse client participation, and stragglers through a simple, loss-based embedding approach.
- **Robustness against initialization**: Unlike existing methods, *CLoVE* does not require careful initialization of cluster models, thereby mitigating the risk of inaccurate cluster assignments due to initialization sensitivities.
- **Theoretical guarantees for cluster recovery**: We provide analytical results showing that, in the context of linear regression, *CLoVE* can achieve single-shot, accurate cluster recovery with high probability.
- **Efficient cluster recovery and model construction**: We demonstrate through comprehensive evaluations that *CLoVE* can recover the true clusters with high accuracy and construct per-cluster models within a few rounds of federated learning in a variety of settings.

## 2 RELATED WORK

A primary challenge in FL is handling heterogeneous data distributions (non-IID data) and varying device capabilities. Early optimization methods like *FedAvg* (McMahan et al., 2017) and *FedProx* (Li et al., 2019) mitigate some issues by allowing multiple local computations before aggregation, yet they typically yield a single global model that may not capture the diversity across user populations (Zhao et al., 2018).

Personalized Federated Learning (PFL) (Hanzely & Richtárik, 2020; Tan et al., 2023) aims to deliver client-specific models that outperform both global models and naïve local baselines. Various strategies have been proposed, including fine-tuning global models locally (Fallah et al., 2020) or performing adaptive local aggregation (Zhang et al., 2023), meta-learning (Jiang et al., 2023), multi-task learning (Smith et al., 2017; Xu et al., 2023), and decomposing models into global and personalized components (Deng et al., 2020; Mansour et al., 2020b). However, these methods operate in a supervised setting and require labeled data.

Clustered Federated Learning (CFL) is an emerging approach in PFL that involves clustering clients based on similarity measures to train specialized models for each cluster (e.g. *IFCA* based on model loss values, (Werner et al., 2023; Kim et al., 2024) and *FlexCFL* based on gradients). The *CFL-S* algorithm utilizes federated multitask learning and gradient embeddings in order to iteratively form and refine clusters. Vahidian et al. (2023)'s *PACFL* algorithm derives a set of principal vectors from each cluster's data, and *FedProto* (Tan et al., 2022) constructs class prototypes, which allow the server to identify distribution similarities.

Traditional clustering methods, such as spectral clustering and k-means, have also been adapted for federated scenarios. In *FeSEM*, personalization is achieved by deriving the optimal matching of users and multiple model centers, and in Chen et al. (2023) through spectral co-distillation. Licciardi et al. (2025) is a hierarchical approach like Sattler et al. (2021), and as such the number of rounds for clustering can be very large. Despite their effectiveness, these methods often require complex similarity computations, additional communication steps, or additional amount of state kept at the server.

Recent advancements in CFL have focused on enhancing convergence and reducing reliance on initial cluster assignments. Improved algorithms (Vardhan et al., 2024) address some limitations of earlier approaches that require careful initialization like *IFCA*, but their methods are complex. Our *CLoVE* algorithm overcomes these drawbacks using a simple method that utilizes robust loss-based

embeddings, converges quickly to the correct clusters, eliminates the need for precise initialization, and operates in a variety of both supervised and unsupervised settings.

# 3 EMBEDDINGS-DRIVEN FEDERATED CLUSTERING

The typical FL architecture consists of a server and a number of clients. Let $M$ be the number of clients, which belong to $K$ clusters. Data for clients of cluster $j \in [K]$ follow a distinct data distribution $D_j$ (where the notation $[K]$ denotes the set $\{1, 2, ..., K\}$). Each client $i \in [M]$ has $n_i$ data points. Let $f(\theta; z) : \Theta \to \mathbb{R}$ be the loss function associated with data point $z$, where $\Theta \subseteq \mathbb{R}^d$ is the parameter space. The goal in CFL is to simultaneously cluster the clients into $K$ clusters and train $K$ models to minimize the population loss function $\mathcal{L}_j(\theta) \triangleq \mathbb{E}_{z \sim D_j}[f(\theta; z)]$ for each cluster $j \in [K]$. The empirical loss for the data $Z_i$ of client $i$ is defined as $\mathcal{L}^i(\theta; Z_i) \triangleq (1/|Z_i|) \sum_{z \in Z_i} f(\theta; z)$.

---

**Algorithm 1:** Clustering of Loss Vector Embeddings (*CLoVE*) algorithm for PFL

---

**Input:** # of clients $M$, upper bound on # of models $\hat{K}$, # of data points $n_i$ for client $i$, $i \in [M]$, participation rate $\rho$

**Hyperparameters:** learning rate $\gamma$, # of rounds $T$, # of local epochs $\tau$ (for model averaging)

**Output:** number of clusters $K^{(T)}$, final model parameters $\left\{ \theta_j^{(T)}, j \in [K^{(T)}] \right\}$

---

1   Server: initialize model parameters $\theta_j^{(0)}$ for each of the $K^{(0)}$ models $j \in [K^{(0)}]$, where $K^{(0)} = \hat{K}$

2   **for** *round* $t = 0, 1, ..., T - 1$ **do**

3     $M^{(t)} \leftarrow$ random subset of $\rho$ fraction of the clients (deals with partial participation/stragglers)

4     **if** *clustering has not stabilized* **then**

5       Server: broadcast model parameters $\theta_j^{(t)}$ of all $\hat{K}$ models $j \in [\hat{K}]$ to all clients in $M^{(t)}$

6       **for** *each client* $i \in M^{(t)}$ *in parallel* **do**

7         Run client $i$'s data through all $\hat{K}$ models and get the losses $\mathcal{L}^{i,(t)} \triangleq \left( \mathcal{L}^i(\theta_j^{(t)}) \right), j \in [\hat{K}]$

8         Send loss vector $\mathcal{L}^{i,(t)}$ to the server

9       Server (clustering step):

10       – $K^{(t+1)} = selectBestK(\mathcal{L}^{(t)}, \hat{K})$, where $\mathcal{L}^{(t)} \triangleq \{\mathcal{L}^{i,(t)}, i \in [M^{(t)}]\}$ is the loss matrix

11       – Run clustering alg. to group the set of vectors $\{\mathcal{L}^{i,(t)}, i \in [M^{(t)}]\}$ to form $K^{(t+1)}$ client clusters

12       – Map client clusters to models using min-cost matching and assign each client $i \in M^{(t)}$ to the corresponding model $\kappa_i \equiv \kappa_i^{(t)} \in [K^{(t+1)}]$ of its client cluster

13     **else**

14       Server:

15       For new clients: assign them a model using models and centroids saved at stability

16       Send updated model parameters of its assigned model $\kappa_i$ to each client $i \in M^{(t)}$

17     **for** *each client* $i \in M^{(t)}$ *in parallel* **do**

18       – **Option I** (model averaging): Compute new model parameters $\widetilde{\theta}_i = \texttt{LocalUpdate}(i, \theta_{\kappa_i}^{(t)}, \gamma, \tau)$ for model $\kappa_i$ and send them to the server

19       – **Option II** (gradient averaging): Compute stochastic gradient $g_i = \widehat{\nabla} \mathcal{L}^i(\theta_{\kappa_i}^{(t)})$ using model $\kappa_i$ and send it to the server

20     Server (averaging step):

21     Clients that were assigned to model $j$ at round $t$: $C_j^{(t)} \triangleq \{i \in M^{(t)} \text{ s.t. } \kappa_i^{(t)} = j\}$

22     – **Option I** (model averaging): $\forall j \in [K^{(t+1)}]$ s.t. $C_j^{(t)} \neq \emptyset$, $\theta_j^{(t+1)} = \dfrac{\sum_{i \in C_j^{(t)}} n_i \widetilde{\theta}_i}{\sum_{i \in C_j^{(t)}} n_i}$

23     – **Option II** (gradient averaging): $\forall j \in [K^{(t+1)}]$ s.t. $C_j^{(t)} \neq \emptyset$, $\theta_j^{(t+1)} = \theta_j^{(t)} - \gamma \dfrac{\sum_{i \in C_j^{(t)}} n_i g_i}{\sum_{i \in C_j^{(t)}} n_i}$

24   **return** $K^{(T)}, \left\{ \theta_j^{(T)}, j \in [K^{(T)}] \right\}$

---

The pseudocode of our *Clustering of Loss Vector Embeddings* (*CLoVE*) PFL algorithm is shown in Alg. 1. The algorithm operates as follows. Initially, the server creates $\hat{K}$ models, where $\hat{K}$ is the upper bound on the number of models, given as an input (line 1). Until cluster stability is reached, it

sends all $\hat{K}$ models to a randomly selected fraction $\rho$ of the participating clients in round $t$ (lines 3, 5). Each of these clients runs its data through all models and gets the losses, which form one loss vector of length $\hat{K}$ per client. These loss vectors are then sent to the server (lines 6-8). The server, if needed, determines the number of clusters $K^{(t+1)}$ for the next round (e.g. via searching over a range of values and choosing the one that yields the highest silhouette score, or using the elbow method) by calling Alg. 2. It then groups clients into $K^{(t+1)}$ clusters (line 11) by clustering their loss vectors (e.g. using k-means). It then assigns clients to those models to which their clusters get matched via a minimum-cost matching (line 12). For this, a bipartite graph is created, with the nodes being the client clusters on the left and the models on the right, and the weight of each edge being the sum of the losses of the clients in the cluster on the left on the model on the right. Min-cost bipartite matching is found using well known methods (e.g. max-flow-based (Munkres, 1957; Lovász & Plummer, 2009); (Jonker & Volgenant, 1987)). Subsequently, each client trains its assigned model $\kappa_i$ locally on its private data (lines 17-19) by running gradient descent denoted by the function LocalUpdate($\cdot$) for $\tau$ epochs, starting with model $\kappa_i$'s parameters, and sends its updates (model parameters in Option 1 and gradients in Option II) to the server. Then, the averaging step follows: the server aggregates and applies all the received updates for each model $j$ from the set $C_j^{(t)}$ of clients that updated it (lines 20-23). In the next iteration, the server sends the computed updated parameters once again to the clients (line 5), and the process continues, until the clustering (i.e. assignment of clients to clusters/models) stabilizes.

---

**Algorithm 2:** Selection of the number of clusters

**Input:** Loss matrix $\mathcal{L}$, initial # of models $K^{(0)}$
**Output:** Best number of clusters $K_{best}$

1  selectBestK($\mathcal{L}$, $K^{(0)}$)
2  $best\_score = -1$
3  **for** $K = 2, ..., K^{(0)}$ **do**
4      $Z_K =$ Clustering of the loss vectors (rows of $\mathcal{L}$) using a clustering algorithm, e.g. Agglomerative Clustering, into $K$ clusters
5      $score = silhouette\_score(\mathcal{L}, Z_K)$
6      **if** $score > best\_score$ **then**
7          $best\_score = score$
8          $K_{best} = K$
9  **return** $K_{best}$

---

Checking for cluster stability can be done as follows, in a way that the server does not need to keep any client-specific state. Each client can keep the history of its assignment to models over the different rounds. Once a client determines that its historical assignments have been stable for a set number of rounds required for stability, the client declares to the server that it has reached stability. The server at each round checks whether all (or a desired percentage) of the participating clients have reached stability, and if so, it decides that universal stability has been achieved.

After the algorithm reaches a stable state, the server no longer needs to compute loss vectors or perform clustering. Instead, it can simply send the already assigned (single) model to each client (line 16). For any client that does not yet have an assigned model, the server assigns it the model whose cluster centroid is closest to the client's loss vector using the models and centroids saved at the point of stability (line 15). At this stage, the algorithm behaves like standard *FedAvg*, so both the communication and the computational overhead are minimal.

If the number of clusters is unknown a priori, then the choice of $K^{(t+1)}$ at each round happens as follows, as per Alg. 2. A range of candidate $K$ values is examined, up to a predetermined upper bound $\hat{K}$. For each candidate $K$, the loss vectors are clustered and the silhouette score of the resulting clustering is computed. Eventually, the value of $K$ with the highest silhouette score is returned. Note that this procedure is not necessary when the number of clusters $K$ is known a priori.

## 4 THEORETICAL ANALYSIS OF *CLoVE*'S PERFORMANCE

We now provide theoretical evidence supporting the superior performance of *CLoVE*. In the following we omit all proofs, which can be found in Appendix A. To analyze the efficacy of *CLoVE*, we consider

a mixed linear regression problem (Yi et al., 2014) in a federated learning setting (Ghosh et al., 2022). In this context, each client's data originates from one of $K$ (where $K$ is fixed and known a priori) distinct clusters, denoted as $\mathcal{C}_1, \mathcal{C}_2, \ldots, \mathcal{C}_K$. Within each cluster $\mathcal{C}_k$, the data pairs $z_i = (x_i, y_i)$ are drawn from the same underlying distribution $\mathcal{D}_k$, which is generated by the linear model:

$$y_i = \langle \theta_k^*, x_i \rangle + \epsilon_i.$$

Here, the feature $x_i$ follows a standard normal distribution $\mathcal{N}(0, I_d)$, and the additive noise $\epsilon_i$ follows a normal distribution $\mathcal{N}(0, \sigma^2)$ ($\sigma \ll 1$), both independently drawn. The loss function is the squared error: $f(\theta; x, y) = (\langle \theta, x \rangle - y)^2$. Under this setting, the parameters $\theta_k^* \in \mathbb{R}^d$ are the minimizers of the population loss for cluster $\mathcal{C}_k$.

At a high level, our analysis unfolds as follows. We demonstrate that by setting the entries of a client's loss vector (SLV) to the square root of model losses, *CLoVE* can accurately recover client clustering with high probability in each round. This, in turn, enables *CLoVE* to accurately match clients to models with high probability by utilizing the minimum-cost bipartite matching between clusters and models. Consequently, we show that the distance between the models constructed by *CLoVE* and their optimal counterparts decreases exponentially with the number of rounds, following the application of client gradient updates, thus implying fast convergence of *CLoVE* to optimal models.

For our analysis we make some assumptions, including that all $\theta_i^* \in \mathbb{R}^d$ have unit norms and their minimum separation $\Delta > 0$ is close to 1. We also assume the number of clusters $K$ is small in comparison to the total number of clients $M$ and the dimension $d$. We assume that each client of cluster $k$ uses the same number $n_k$ of i.i.d. data points independently chosen at each round.

For a given model $\theta$, the **empirical loss** $\mathcal{L}_k^i(\theta)$ for a client $i$ in cluster $k$ is:

$$\frac{1}{n_k} \sum_{j=1}^{n_k} (\langle \theta, x_j \rangle - y_j)^2 = \frac{1}{n_k} \sum_{j=1}^{n_k} (\langle \theta_k^* - \theta, x_j \rangle + \epsilon_j)^2$$

For a given collection of models, $\boldsymbol{\theta} = [\theta_1, \theta_2, \cdots]$, the square root loss vector (SLV) of a client $i$ of cluster $k$ is $a_k^i(\boldsymbol{\theta}) = [F_k^i(\theta_1), F_k^i(\theta_2), \ldots]$. Here $F_k^i(\theta) = \sqrt{\mathcal{L}_k^i(\theta)} \sim \frac{\alpha(\theta, \theta_k^*)}{\sqrt{n_k}} \chi(n_k)$, where $\alpha(\theta, \theta_k^*) = \sqrt{\|\theta - \theta_k^*\|^2 + \sigma^2}$, and $\chi(n_k)$ denotes a standard chi random variable with $n_k$ degrees of freedom. Let $c_k(\boldsymbol{\theta}) = [\mathbb{E}[F_k^i(\theta_1)], \mathbb{E}[F_k^i(\theta_2)], \ldots]$ denote the distribution mean of the SLVs for a cluster $k$. In the following $m_k$ denotes the number of clients in cluster $k$ and $n = \min_k n_k$.

We say a collection of $K$ models $\boldsymbol{\theta} = [\theta_1, \theta_2, \cdots \theta_K]$ is $\Delta$-proximal if each $\theta_k$ is within a distance at most $\Delta/4$ of its optimal counterpart $\theta_k^*$. For our analysis we assume *CLoVE* is initialized with a set $\boldsymbol{\theta}$ of $d$ randomly drawn (hence nearly ortho-normal) models. One of our key results is the following.

**Theorem 4.1.** *For both ortho-normal or $\Delta$-proximal models collections, k-means clustering of SLVs, with suitably initialized centers, recovers accurate clustering of the clients in one shot. This result holds with probability at least $\xi = 1 - \delta - \frac{1}{polylog(M)}$, for any error tolerance $\delta$.*

We prove this by showing that, for such model collections, the mixture of distributions formed by the collection of SLVs from different clients satisfies the proximity condition of Kumar & Kannan (2010). Specifically, it is shown in Theorem A.9 that when the number of data points across all clients is much larger than $d, K$, then for any client $i$, with probability at least $\xi$:

$$\forall\, j' \neq j, \ \|a_j^i(\boldsymbol{\theta}) - c_{j'}(\boldsymbol{\theta})\| - \|a_j^i(\boldsymbol{\theta}) - c_j(\boldsymbol{\theta})\| \geq \left( \frac{c'K}{m_i} + \frac{c'K}{m_j} \right) \|A(\boldsymbol{\theta}) - C(\boldsymbol{\theta})\|$$

Here $c'$ is a large constant and client $i$ is assumed to belong to cluster $j$. Theorem 4.1 then follows directly from the result of Kumar & Kannan (2010).

The proof of Theorem A.9 is based on a sequence of intermediate results. In particular, Lemma A.1 and Lemma A.2 demonstrate that, for the ortho-normal and $\Delta$-proximal model collections, the means of the SLV distributions across distinct clusters are sufficiently well-separated:

For any $i \neq j$, $\|c_i(\boldsymbol{\theta}) - c_j(\boldsymbol{\theta})\| \geq \frac{\Delta}{c}$. Here, $c \approx 2$.

Furthermore, Lemma A.4 shows that with high probability, SLVs of each cluster are concentrated around their means. We use these results, combined with a result of Dasgupta et al. (2007) to prove

Lemma A.7 which establishes that the norm of the matrix $X(\boldsymbol{\theta})$ of "centered SLVs", whose $i$-th row is $a_{s(i)}^i(\boldsymbol{\theta}) - c_{s(i)}(\boldsymbol{\theta})$, is bounded. That is, for $\gamma = \sqrt{d\frac{1}{2n}(9 + \sigma^2)}$, with high probability: $\|X(\boldsymbol{\theta})\| \leq \gamma\sqrt{M}\mathrm{polylog}(M)$. By combining these results, we are able to derive the proximity condition of Theorem A.9.

Next, we prove our second key result, that models constructed by *CLoVE* converge exponentially fast towards their optimal counterpart. Let $\boldsymbol{\theta}^i$ denote model collection at round $i$.

**Theorem 4.2.** *After $T$ rounds, each model $\theta_k \in \boldsymbol{\theta}^T$ satisfies $\|\theta_k - \theta_k^*\| \leq \frac{\Delta}{c_1^T}$, with high probability, for a constant $c_1$.*

In the first round, the models $\boldsymbol{\theta}^1$ for *CLoVE* are initialized to be ortho-normal, ensuring accurate cluster recovery by Theorem 4.1. Under the assumption that for all clusters $k$, $n_k m_k \left(\frac{\rho}{\rho+1}\right)^2 \gg \mathrm{polylog}(K)$, where $\rho = \Delta^2/\sigma^2$, Lemma A.16 shows that the min-cost bi-bipartite matching found by *CLoVE* yields an optimal client-to-model match with probability close to 1. Furthermore, Lemma A.13 demonstrates that as a consequence of this optimal match, the updated $K$ models $\boldsymbol{\theta}^2$, obtained by applying gradient updates from clients to their assigned models, are $\Delta$-proximal with high probability, provided that the learning rate $\eta$ is suitably chosen.

In subsequent rounds, the $\Delta$-proximality of the models $\boldsymbol{\theta}^2$ enables the application of a similar argument, augmented by Corollary A.14. With high probability, this implies that the updated $K$ models $\boldsymbol{\theta}^3$ not only remain $\Delta$-proximal, but also have a distance to their optimal counterparts that is a constant fraction $c_1$ times smaller, resulting from the gradient updates from their assigned clients. Thus, by induction, the proof of Theorem 4.2 follows, provided $\delta T \ll 1$.

## 5 Performance Evaluation

We now show how our algorithms compare to the state of the art in both supervised and unsupervised settings. In the unsupervised setting in particular, which is understudied in the context of CFL, our evaluation is, to the best of our knowledge, the first extensive one. The datasets, models and data distributions we use follow the ones used by the state of the art (Vahidian et al., 2023; Duan et al., 2022; Xu et al., 2023; Zhang et al., 2023; Ghosh et al., 2020).

### 5.1 Experimental Setup

**Datasets and Models**   We evaluate our method on three types of tasks – image classification, text classification, and image reconstruction – using six widely-used datasets: MNIST, Fashion-MNIST (FMNIST), CIFAR-10, FEMNIST, Amazon Reviews, and AG News. For the classification tasks, we use three different convolutional neural networks (CNNs): a shared CNN model for MNIST, FMNIST and FEMNIST, a deeper CNN for CIFAR-10, a TextCNN for AG News, and an MLP-based model (AmazonMLP) for Amazon Reviews. For the unsupervised image reconstruction tasks on MNIST and FMNIST, we use a simple autoencoder, while a convolutional autoencoder is employed for CIFAR-10. Further details about these datasets and model architectures can be found in App. C.

**Dataset Partitioning**   As in the state of the art, we extensively cover many high variance and high overlap data heterogeneity scenarios through random data partitioning, without replacement, with various types of label and feature skews, as well as concept shifts. For label skew, we consider many non-overlapping and overlapping label distributions. We present only some of the results here (rest are in App. D.1.2 and D.1.3). In case *Label skew 1*, clients within each cluster receive data from only two unique classes, with no label overlap between clients in different clusters. Conversely, in case *Label skew 2*, each client within each cluster receives samples from $U$ classes, with at least $V$ labels shared between the labels assigned to any two clusters. For the AG News dataset, we use $U = 2$ and $V = 1$, while for all image datasets we use $U = 4$ and $V = 2$. As is standard practice in the FL literature, for *Label skew 1* and *2* (and *3* in App. D.1.2), the data of a class is distributed amongst the clients that are assigned to that class by sampling from a Dirichlet distribution with parameter $\alpha = 0.5$. For instance, suppose the clients are divided into three clusters, and a particular class contains $L$ data points. First, we draw three proportions $p_1, p_2, p_3$ from a Dirichlet distribution with concentration parameter $\alpha = 0.5$ (by construction, $p_1 + p_2 + p_3 = 1$). Then we randomly select $Lp_1$ points of the class for the clients in cluster 1, $Lp_2$ points for the clients in cluster 2, and the

remaining $Lp_3$ points for the clients in cluster 3. The selected points are assigned randomly among the clients within each respective cluster.

To simulate feature distribution shifts, we apply image rotations (0°, 90°, 180°, 270°) to the MNIST, CIFAR-10 and FMNIST datasets. Each client within a cluster is assigned data with a specific rotation. Concept shift is achieved through label permutation: all labels are distributed across all clusters, but clients within each cluster receive data with two unique label swaps. Both test and train data of a client have the same distribution. No data mixing is employed for the Amazon Review dataset, i.e. we use its original data classes.

**Compared Methods**  We compare the following baselines: *Local-only*, where each client trains its model locally; *FedAvg*, which learns a single global model; clustered federated learning methods, including *CFL-S*, *IFCA*, *FlexCFL*, *FeSEM*, and *PACFL* (Vahidian et al., 2023); and general PFL methods *FedProto* (Tan et al., 2022), *Per-FedAvg* (Fallah et al., 2020), *FedALA* (Zhang et al., 2023), and *FedPAC* (Xu et al., 2023). The source code we used for these algorithms is linked in Table 7.

**Training Settings**  We use an Adam (Kingma & Ba, 2017) optimizer, a batch size of 64 and a learning rate $\gamma = 10^{-3}$. The number of local training epochs is $\tau = 1$ and the number of global communication rounds is $T = 100$. For supervised classification tasks we use cross-entropy loss, and for unsupervised tasks we use the MSE reconstruction loss. We vary the number of clients between 20 and 1000. In most cases, the training data size per client is set to 500. However, for the Amazon Reviews and AG News datasets, we use 1000 and 100 samples, respectively. Details on the number of clients and training data sizes for all experiments is provided in App. C.

**Result reporting**  We report the average performance of the models assigned to the clients on their local test data. Specifically, we report test accuracies for supervised classification tasks and reconstruction losses for unsupervised tasks. In addition, we report the accuracy of client-to-cluster assignments as the Adjusted Rand Index (ARI) between the achieved clustering and the ground-truth client groupings. To account for randomness, we run each experiment for 3 values of a randomness seed and report the mean and standard deviation. In this section, we only report results for full participation, known number of clusters, and no early stopping of clustering. The results for the other

Table 1: Supervised test accuracy results for MNIST, CIFAR-10 and FMNIST

| Algorithm | Data mixing | MNIST | CIFAR-10 | FMNIST | Data mixing | MNIST | CIFAR-10 | FMNIST |
|---|---|---|---|---|---|---|---|---|
| FedAvg | | $87.4 \pm 0.9$ | $34.0 \pm 3.6$ | $64.7 \pm 1.2$ | | $52.4 \pm 0.3$ | $\mathbf{63.1 \pm 1.7}$ | $49.1 \pm 0.1$ |
| Local-only | | $99.5 \pm 0.1$ | $85.4 \pm 1.6$ | $98.8 \pm 0.1$ | | $94.8 \pm 0.3$ | $45.1 \pm 0.4$ | $78.3 \pm 0.4$ |
| Per-FedAvg | | $97.8 \pm 0.2$ | $75.0 \pm 0.7$ | $97.0 \pm 0.1$ | | $80.2 \pm 1.0$ | $34.9 \pm 0.7$ | $70.4 \pm 1.0$ |
| FedProto | Label skew 1 | $99.2 \pm 0.0$ | $83.0 \pm 0.3$ | $98.6 \pm 0.0$ | Concept shift | $95.0 \pm 0.4$ | $40.8 \pm 0.3$ | $78.4 \pm 0.3$ |
| FedALA | | $98.9 \pm 0.1$ | $80.0 \pm 0.6$ | $97.8 \pm 0.0$ | | $96.7 \pm 0.2$ | $42.0 \pm 1.0$ | $81.1 \pm 0.3$ |
| FedPAC | | $94.4 \pm 0.7$ | $78.3 \pm 0.4$ | $91.7 \pm 1.2$ | | $90.7 \pm 1.3$ | $39.0 \pm 1.0$ | $73.8 \pm 0.6$ |
| CFL-S | | $82.5 \pm 21.8$ | $87.4 \pm 0.4$ | $65.8 \pm 22.4$ | | $97.5 \pm 0.2$ | $35.7 \pm 3.2$ | $83.7 \pm 0.5$ |
| FeSEM | | $73.2 \pm 0.8$ | $11.2 \pm 1.6$ | $41.8 \pm 0.7$ | | $46.2 \pm 0.1$ | $12.7 \pm 0.5$ | $39.7 \pm 0.6$ |
| FlexCFL | | $99.4 \pm 0.0$ | $73.2 \pm 0.3$ | $99.0 \pm 0.1$ | | $97.4 \pm 0.1$ | $17.8 \pm 2.1$ | $\mathbf{86.1 \pm 0.2}$ |
| PACFL | | $99.1 \pm 0.1$ | $79.1 \pm 1.0$ | $98.7 \pm 0.1$ | | $93.3 \pm 0.1$ | $26.8 \pm 1.2$ | $72.6 \pm 1.2$ |
| IFCA | | $99.1 \pm 0.5$ | $85.7 \pm 6.5$ | $97.8 \pm 1.9$ | | $80.0 \pm 0.2$ | $52.7 \pm 4.7$ | $67.0 \pm 6.2$ |
| CLoVE | | $\mathbf{99.8 \pm 0.0}$ | $\mathbf{90.5 \pm 0.1}$ | $\mathbf{99.1 \pm 0.0}$ | | $\mathbf{97.6 \pm 0.1}$ | $58.4 \pm 0.6$ | $84.3 \pm 0.2$ |
| FedAvg | | $76.4 \pm 0.3$ | $47.7 \pm 4.0$ | $55.2 \pm 1.6$ | | $90.0 \pm 0.6$ | $52.4 \pm 2.5$ | $76.5 \pm 0.3$ |
| Local-only | | $94.2 \pm 0.2$ | $55.9 \pm 1.8$ | $86.1 \pm 0.4$ | | $93.0 \pm 0.1$ | $38.5 \pm 1.6$ | $75.8 \pm 0.2$ |
| Per-FedAvg | | $94.9 \pm 0.6$ | $48.2 \pm 0.9$ | $78.2 \pm 1.0$ | | $87.8 \pm 0.5$ | $22.8 \pm 0.1$ | $66.7 \pm 0.4$ |
| FedProto | Label skew 2 | $94.4 \pm 0.2$ | $56.8 \pm 0.8$ | $82.1 \pm 0.9$ | Feature skew | $92.3 \pm 0.3$ | $34.1 \pm 0.2$ | $74.0 \pm 0.3$ |
| FedALA | | $95.1 \pm 0.3$ | $50.1 \pm 1.6$ | $81.6 \pm 0.7$ | | $90.2 \pm 0.6$ | $22.3 \pm 0.4$ | $68.1 \pm 0.5$ |
| FedPAC | | $91.9 \pm 0.5$ | $40.7 \pm 5.2$ | $77.6 \pm 1.2$ | | $87.6 \pm 0.4$ | $24.6 \pm 0.2$ | $67.4 \pm 0.2$ |
| CFL-S | | $\mathbf{97.3 \pm 0.3}$ | $62.1 \pm 0.4$ | $87.2 \pm 1.2$ | | $95.1 \pm 0.2$ | $\mathbf{61.6 \pm 5.1}$ | $77.8 \pm 3.5$ |
| FeSEM | | $89.1 \pm 1.6$ | $15.3 \pm 2.5$ | $51.4 \pm 0.2$ | | $79.5 \pm 0.4$ | $10.7 \pm 0.5$ | $67.6 \pm 0.5$ |
| FlexCFL | | $95.6 \pm 0.3$ | $43.6 \pm 0.6$ | $88.7 \pm 0.2$ | | $96.7 \pm 0.1$ | $12.2 \pm 0.4$ | $\mathbf{85.2 \pm 0.2}$ |
| PACFL | | $92.0 \pm 1.1$ | $40.4 \pm 1.0$ | $81.1 \pm 0.5$ | | $89.3 \pm 0.1$ | $17.1 \pm 0.6$ | $70.6 \pm 0.6$ |
| IFCA | | $96.8 \pm 0.2$ | $63.4 \pm 3.4$ | $89.4 \pm 1.3$ | | $95.9 \pm 2.0$ | $50.8 \pm 7.0$ | $83.5 \pm 0.3$ |
| CLoVE | | $97.1 \pm 0.3$ | $\mathbf{66.8 \pm 2.3}$ | $\mathbf{90.1 \pm 0.3}$ | | $\mathbf{97.7 \pm 0.1}$ | $57.1 \pm 1.8$ | $85.1 \pm 0.2$ |

Table 2: Supervised ARI results for MNIST, CIFAR-10 and FMNIST

| Algorithm | Data mixing | MNIST | CIFAR-10 | FMNIST | Data mixing | MNIST | CIFAR-10 | FMNIST |
|---|---|---|---|---|---|---|---|---|
| CFL-S | | $0.71 \pm 0.20$ | $\mathbf{1.00 \pm 0.00}$ | $0.86 \pm 0.20$ | | $\mathbf{1.00 \pm 0.00}$ | $0.00 \pm 0.00$ | $\mathbf{1.00 \pm 0.00}$ |
| FeSEM | | $0.18 \pm 0.00$ | $0.00 \pm 0.00$ | $0.00 \pm 0.00$ | | $0.00 \pm 0.00$ | $0.00 \pm 0.00$ | $0.00 \pm 0.00$ |
| FlexCFL | Label skew 1 | $\mathbf{1.00 \pm 0.00}$ | $\mathbf{1.00 \pm 0.00}$ | $\mathbf{1.00 \pm 0.00}$ | Concept shift | $\mathbf{1.00 \pm 0.00}$ | $0.22 \pm 0.24$ | $\mathbf{1.00 \pm 0.00}$ |
| PACFL | | $0.39 \pm 0.07$ | $0.60 \pm 0.09$ | $\mathbf{1.00 \pm 0.00}$ | | $0.00 \pm 0.02$ | $0.00 \pm 0.00$ | $0.00 \pm 0.00$ |
| IFCA | | $0.83 \pm 0.12$ | $0.91 \pm 0.13$ | $0.92 \pm 0.12$ | | $0.68 \pm 0.00$ | $0.76 \pm 0.17$ | $0.55 \pm 0.18$ |
| CLoVE | | $\mathbf{1.00 \pm 0.00}$ | $\mathbf{1.00 \pm 0.00}$ | $\mathbf{1.00 \pm 0.00}$ | | $\mathbf{1.00 \pm 0.00}$ | $\mathbf{1.00 \pm 0.00}$ | $\mathbf{1.00 \pm 0.00}$ |
| CFL-S | | $0.00 \pm 0.00$ | $0.72 \pm 0.00$ | $0.57 \pm 0.00$ | | $0.48 \pm 0.00$ | $0.89 \pm 0.15$ | $0.16 \pm 0.23$ |
| FeSEM | | $0.18 \pm 0.00$ | $0.00 \pm 0.00$ | $0.00 \pm 0.00$ | | $0.00 \pm 0.00$ | $0.00 \pm 0.00$ | $0.00 \pm 0.00$ |
| FlexCFL | Label skew 2 | $\mathbf{1.00 \pm 0.00}$ | $\mathbf{1.00 \pm 0.00}$ | $\mathbf{1.00 \pm 0.00}$ | Feature skew | $\mathbf{1.00 \pm 0.00}$ | $0.11 \pm 0.08$ | $\mathbf{1.00 \pm 0.00}$ |
| PACFL | | $0.35 \pm 0.12$ | $0.32 \pm 0.11$ | $0.55 \pm 0.03$ | | $0.00 \pm 0.00$ | $0.00 \pm 0.00$ | $0.68 \pm 0.11$ |
| IFCA | | $0.83 \pm 0.12$ | $0.81 \pm 0.13$ | $0.92 \pm 0.12$ | | $0.67 \pm 0.28$ | $0.71 \pm 0.22$ | $0.55 \pm 0.10$ |
| CLoVE | | $\mathbf{1.00 \pm 0.00}$ | $\mathbf{1.00 \pm 0.00}$ | $\mathbf{1.00 \pm 0.00}$ | | $\mathbf{1.00 \pm 0.00}$ | $\mathbf{1.00 \pm 0.00}$ | $\mathbf{1.00 \pm 0.00}$ |

cases, as well as scaling experiments and ablation studies, are provided in App. D and show that *CLoVE* performs well even in these settings.

## 5.2 NUMERICAL RESULTS

The test accuracies for image classification tasks are reported in Table 1. The results demonstrate that *CLoVE* performs consistently well across various data distributions, including label and feature skews, as well as concept shifts. While some baseline algorithms exhibit strong performance in specific cases, **CLoVE is the only algorithm that consistently ranks among the top performers in almost all scenarios**.

Table 2 presents the clustering accuracies of different CFL algorithms for the image datasets under the same experimental setup. *CLoVE* achieves optimal performance in all cases. Although *FlexCFL* also shows near-optimal performance, its accuracy declines when faced with concept shifts, especially on CIFAR-10, as it neglects label information in similarity estimation. In contrast, loss-based approaches like *CLoVE* excel in this metric,
as model losses can effectively account for various types of skews. Results for additional types of label skews (e.g. dominant class (Xu et al., 2023)) are included in App. D.1.2.

Table 3: Supervised test accuracy results for Amazon Review and AG News

| Algorithm | Amazon | AG News |
|---|---|---|
| FedAvg | $\mathbf{99.0 \pm 0.1}$ | $40.2 \pm 0.1$ |
| Local-only | $81.2 \pm 0.4$ | $51.4 \pm 1.1$ |
| Per-FedAvg | $87.7 \pm 0.4$ | $48.0 \pm 1.1$ |
| FedProto | $81.3 \pm 0.1$ | $18.9 \pm 1.5$ |
| FedALA | $88.3 \pm 0.3$ | $49.6 \pm 0.1$ |
| FedPAC | $88.2 \pm 0.2$ | $42.4 \pm 2.9$ |
| CFL-S | $87.7 \pm 0.8$ | $48.4 \pm 2.7$ |
| FeSEM | $50.9 \pm 0.2$ | $19.1 \pm 1.5$ |
| FlexCFL | $84.1 \pm 0.8$ | $\mathbf{60.2 \pm 5.5}$ |
| PACFL | $82.1 \pm 0.3$ | $51.1 \pm 1.9$ |
| IFCA | $85.5 \pm 0.5$ | $51.3 \pm 3.3$ |
| CLoVE | $86.8 \pm 0.4$ | $55.1 \pm 0.2$ |

The test accuracies for the textual classification tasks are reported in Table 3 (more metrics in App. D.1.3). While *CLoVE* exhibits good performance here as well, the baseline *FedAvg* performs especially well on the Amazon Review dataset. One reason is that while the dataset is split by product categories, the underlying language and sentiment cues are still fairly similar across all 4 categories and can provide a good signal for classification. On AG News, *CLoVE* outperforms all baselines except *FlexCFL*.

Table 5 presents the clustering recovery speed for the image classification datasets under different client data distributions. The first 3 columns show the ARI reached within 10 rounds for each algorithm. *CLoVE* outperforms all baselines here. The next 3 columns show the first round at which an ARI of 0.9 or higher is achieved (dashes mean 0.9 ARI is never achieved). As can be seen, **CLoVE consistently reaches such high accuracy within 2-3 rounds, unlike any other baseline**.

**Unsupervised Setting** Unlike many of the baselines, *CLoVE* is equally applicable to unsupervised settings, as it relies solely on model losses to identify clients with similar data distributions. Another loss-based approach, *IFCA*, serves as a natural point of comparison in this context. As shown in Table 4, *CLoVE* exhibits consistently good performance compared to *IFCA* and other baselines in unsupervised settings on image reconstruction

Table 4: Unsupervised test loss results for MNIST, CIFAR-10 and FMNIST

| Algorithm | MNIST | CIFAR-10 | FMNIST |
|---|---|---|---|
| FedAvg | $0.032 \pm 0.001$ | $0.026 \pm 0.000$ | $0.030 \pm 0.000$ |
| Local-only | $0.019 \pm 0.000$ | $0.022 \pm 0.000$ | $0.170 \pm 0.000$ |
| IFCA | $0.024 \pm 0.005$ | $\mathbf{0.019 \pm 0.000}$ | $0.020 \pm 0.002$ |
| CLoVE | $\mathbf{0.014 \pm 0.000}$ | $0.021 \pm 0.000$ | $\mathbf{0.016 \pm 0.000}$ |

tasks using autoencoders, further highlighting its versatility and effectiveness. More experiments in an unsupervised setting can be found in App. D.2.

**Robustness to initialization** Our results indicate that one key reason *CLoVE* outperforms related baselines such as *IFCA* is its robustness to initialization. We evaluate this by varying two factors: the initialization of model parameters and the first round client-to-model assignment strategy. For model initialization, we consider two settings: (1) all models begin with identical parameters (init:s – same), and (2) each model is initialized independently using PyTorch's default random initialization (init:d – different). For the first round client-to-model assignment, we also explore two approaches: (1) assigning clients to models at random (first:r – random), and (2) assigning clients by clustering their loss vectors (first:e – evaluation-based), followed by bipartite matching as in Alg. 1. Fig. 2 compares *CLoVE* and *IFCA* on unsupervised MNIST image reconstruction with 10 clusters, each containing 5 clients with 500 samples. The results show that *CLoVE* maintains high performance across different initialization methods, unlike *IFCA*.

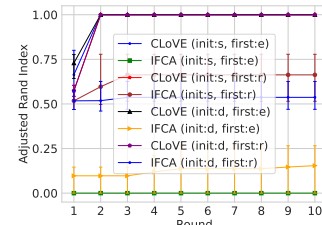

Figure 2: Clustering accuracy of *CLoVE* and *IFCA* over time for the initialization experiments

Table 5: Convergence behavior for MNIST, CIFAR-10 and FMNIST

| Data mixing | Algorithm | ARI reached in 10 rounds | | | First round when ARI $\geq 0.9$ | | |
|---|---|---|---|---|---|---|---|
| | | MNIST | CIFAR-10 | FMNIST | MNIST | CIFAR-10 | FMNIST |
| Label skew 1 | CFL-S | $0.00 \pm 0.00$ | $0.00 \pm 0.00$ | $0.00 \pm 0.00$ | — | $41.3 \pm 2.1$ | — |
| | FeSEM | $0.18 \pm 0.00$ | $0.00 \pm 0.00$ | $0.00 \pm 0.00$ | — | — | — |
| | FlexCFL | $\mathbf{1.00 \pm 0.00}$ | $\mathbf{1.00 \pm 0.00}$ | $\mathbf{1.00 \pm 0.00}$ | $\mathbf{1.0 \pm 0.0}$ | $\mathbf{1.0 \pm 0.0}$ | $\mathbf{1.0 \pm 0.0}$ |
| | PACFL | $0.39 \pm 0.07$ | $0.60 \pm 0.09$ | $\mathbf{1.00 \pm 0.00}$ | — | — | $\mathbf{1.0 \pm 0.0}$ |
| | IFCA | $0.83 \pm 0.12$ | $0.72 \pm 0.00$ | $0.83 \pm 0.12$ | — | — | — |
| | CLoVE | $\mathbf{1.00 \pm 0.00}$ | $\mathbf{1.00 \pm 0.00}$ | $\mathbf{1.00 \pm 0.00}$ | $2.0 \pm 0.0$ | $2.0 \pm 0.0$ | $2.0 \pm 0.0$ |
| Concept shift | CFL-S | $0.00 \pm 0.00$ | $0.00 \pm 0.00$ | $0.00 \pm 0.00$ | $60.0 \pm 9.4$ | — | $43.0 \pm 5.7$ |
| | FeSEM | $0.00 \pm 0.00$ | $0.00 \pm 0.00$ | $0.00 \pm 0.00$ | — | — | — |
| | FlexCFL | $\mathbf{1.00 \pm 0.00}$ | $0.22 \pm 0.24$ | $\mathbf{1.00 \pm 0.00}$ | $\mathbf{1.0 \pm 0.0}$ | — | $\mathbf{1.0 \pm 0.0}$ |
| | PACFL | $0.00 \pm 0.02$ | $0.00 \pm 0.00$ | $0.00 \pm 0.00$ | — | — | — |
| | IFCA | $0.68 \pm 0.00$ | $0.76 \pm 0.17$ | $0.55 \pm 0.18$ | — | — | — |
| | CLoVE | $\mathbf{1.00 \pm 0.00}$ | $\mathbf{1.00 \pm 0.00}$ | $\mathbf{1.00 \pm 0.00}$ | $2.0 \pm 0.0$ | $\mathbf{2.3 \pm 0.5}$ | $2.0 \pm 0.0$ |

## 6 CONCLUSION

We introduced *CLoVE*, a simple loss vector embeddings-based framework for personalized clustered federated learning with low communication and computation overhead that avoids stringent model initialization assumptions and substantially outperforms the state of the art across a range of datasets and in a variety of both supervised and unsupervised settings. Further discussion of the overheads and the privacy properties of *CLoVE* is provided in App. B. In the future, we plan to further explore privacy, as well as fairness and adversarial behavior aspects. Overall, *CLoVE*'s design offers a promising and robust approach to scalable model personalization and clustering under heterogeneous data distributions.

## 7 REPRODUCIBILITY STATEMENT

Our code, together with detailed instructions on how to reproduce our experiments, will be open-sourced. For the purposes of the reviewing process, after the discussion forums open, we will make a comment directed to the reviewers and area chairs and put a link to an anonymous repository containing our code, as mentioned in the Author Guide webpage of the conference (`https://iclr.cc/Conferences/2026/AuthorGuide`). Proofs for our theoretical claims are in App. A. All datasets we are using are already public. We have provided detailed implementation details in Section 5 of the main paper and Appendices C and D.

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

# A  THEORETICAL ANALYSIS

## A.1  BACKGROUND

We consider a mixed linear regression problem (Yi et al., 2014) in a federated setting (Ghosh et al., 2022). Client's data come from one of $K$ different clusters, denoted as $\mathcal{C}_1, \mathcal{C}_2, \ldots, \mathcal{C}_K$. Cluster $k$'s optimal model $\theta_k^* \in \mathbb{R}^d$ is a vector of dimension $d$.

**Assumptions:** We assume all clients of a cluster $\mathcal{C}_k$ use the same number $n_k$ of independently drawn feature-response pairs $(x_i, y_i)$ in each federated round. These feature-response pairs are drawn from a distribution $\mathcal{D}_k$, generated by the model:

$$y_i = \langle \theta_k^*, x_i \rangle + \epsilon_i,$$

where $x_i \sim \mathcal{N}(0, I_d)$ are the features, and $\epsilon_i \sim \mathcal{N}(0, \sigma^2)$ represents additive noise. Both are independently drawn. We assume $\sigma$ is very small. Specifically, $\sigma \ll 1$.

We use notation $[K]$ for the set $\{1, 2, ..., K\}$.

The loss function $f(\theta)$ is defined as the square of the error:

$$f(\theta; x, y) = \left( \langle \theta, x \rangle - y \right)^2.$$

The **population loss** for cluster $k$ is:

$$\mathcal{L}_k(\theta) = \mathbb{E}_{(x,y) \sim \mathcal{D}_k} \left[ \left( \langle \theta, x \rangle - y \right)^2 \right],$$

Note that the optimal parameters $\{\theta_k^*\}_{k=1}^K$ are the minimizers of the population losses $\{\mathcal{L}_k\}_{k=1}^K$, i.e.,

$$\theta_k^* = \arg\min_\theta \mathcal{L}_k(\theta) \quad \text{for each } k \in [K].$$

We assume these optimal models have unit norm. That is $\|\theta_k^*\| = 1$ for all $k$. We also assume the optimal models are well-separated. That is, there exists a $\Delta > 0$ such that for any pair $i \neq j$:

$$\|\theta_i^* - \theta_j^*\| \geq \Delta$$

For a given $\theta$, the **empirical loss** for client $i$ in cluster $k$ is:

$$\mathcal{L}_k^i(\theta) = \frac{1}{n_k} \sum_{j=1}^{n_k} \left( \langle \theta, x_j \rangle - y_j \right)^2 = \frac{1}{n_k} \sum_{j=1}^{n_k} \left( \langle \theta_k^* - \theta, x_j \rangle + \epsilon_j \right)^2$$

Let

$$\alpha(\theta, \theta_k^*) = \sqrt{\|\theta - \theta_k^*\|^2 + \sigma^2}.$$

Note $\mathcal{L}_k^i(\theta) \sim \frac{\alpha(\theta, \theta_k^*)^2}{n_k} \chi^2(n_k)$. That is, it is distributed as a scaled chi-squared random variable with $n_k$ degrees of freedom, with scaling factor $\frac{\alpha(\theta, \theta_k^*)^2}{n_k}$. In the following, we denote the square root of the empirical loss by $F_k^i(\theta)$. That is, $F_k^i(\theta) = \sqrt{\mathcal{L}_k^i(\theta)}$. Note:

$$F_k^i(\theta) \sim \frac{\alpha(\theta, \theta_k^*)}{\sqrt{n_k}} \chi(n_k),$$

where $\chi(n_k)$ denotes a chi-distributed random variable with $n_k$ degrees of freedom.

Using Sterling approximation, $\mathbb{E}[\chi(n)] = \sqrt{n-1} \left[ 1 - \frac{1}{4n} + O\left(\frac{1}{n^2}\right) \right]$. Also, $\text{Var}[\chi(n)] = (n - 1) - \mathbb{E}[\chi(n)]^2 = \frac{n-1}{2n} \left[ 1 + O(\frac{1}{n}) \right]$. Therefore:

$$\mathbb{E}\left[ F_k^i(\theta) \right] = \alpha \left[ 1 - \frac{1}{4n_k} + O\left(\frac{1}{n_k^2}\right) \right] \approx \alpha(\theta, \theta_k^*).$$

$$\text{Var}(F_k^i(\theta)) = \frac{\alpha^2}{n_k} \left( \frac{n_k - 1}{2n_k} + O\left(\frac{1}{n_k}\right) \right) \approx \frac{\alpha(\theta, \theta_k^*)^2}{2n_k}.$$

For $N$ models $\boldsymbol{\theta} = [\theta_1 \quad \theta_2 \quad \cdots \quad \theta_N]$, the square root loss vector (SLV) of a client $i$ of cluster $k$ is $a_k^i(\boldsymbol{\theta}) = [F_k^i(\theta_1), F_k^i(\theta_2), \ldots F_k^i(\theta_N)]$. Let $c_k(\boldsymbol{\theta}) = [c_k^1, c_k^2, \ldots c_k^N]$ be the mean of $k$-th cluster's SLVs. Note that $c_k^j = \mathbb{E}[F_k^i(\theta_j)] = \alpha(\theta_j, \theta_k^*) = \sqrt{\|\theta_j - \theta_k^*\|^2 + \sigma^2}$.

All the model collections $\boldsymbol{\theta}$ considered in this work are assumed to be one of two different types: a) $N = d$ ortho-normal models, such as $d$ models drawn randomly from a $d$-dimensional unit sphere, since such vectors are likely to be orthogonal, and b) $N = K$, $\Delta$-proximal models in which each $\theta_k$ is within a distance at most $\Delta/4$ of its optimal counterpart $\theta_k^*$. Furthermore, the norm of any model in these model collections is required to be bounded. Specifically, $\|\theta\| \leq 2$ for any model $\theta \in \boldsymbol{\theta}$.

## A.2 RESULTS

We prove that the mean of different cluster SLVs are well-separated for our choice of model collections. We first show this for a $N = d$ ortho-normal $\boldsymbol{\theta}$ model collection.

**Lemma A.1.** *For ortho-normal models $\boldsymbol{\theta}$, for any $i \neq j$, $\|c_i(\boldsymbol{\theta}) - c_j(\boldsymbol{\theta})\| \geq \dfrac{\Delta}{c}$, where $c \approx 2$.*

*Proof.* First, we show the result for the following ortho-normal models:

$$\theta_i = (0, 0, \ldots, 1, \ldots, 0) \quad \text{with the 1 in the } i\text{-th position.}$$

Note,

$$\left| \|\theta_i^* - \theta_k\|^2 - \|\theta_j^* - \theta_k\|^2 \right| = 2 \left| \langle (\theta_i^* - \theta_j^*), \theta_k \rangle \right| = 2 |t_k|,$$

where $t_k$ is the $k$-th component of $\theta_i^* - \theta_j^*$. Thus

$$\left| (c_k^i)^2 - (c_k^j)^2 \right| = \left| \|\theta_i^* - \theta_k\|^2 - \|\theta_j^* - \theta_k\|^2 \right| = 2 |t_k|.$$

Also, since $\forall_i, \|\theta_i^*\| = 1$ and $\forall_k, \|\theta_k\| = 1$, it follows for any $k, i$ that $c_k^i = \sqrt{\|\theta_k - \theta_i^*\|^2 + \sigma^2} \leq \sqrt{4 + \sigma^2}$. Thus,

$$\left| c_k^i - c_k^j \right| \geq \frac{|t_k|}{\sqrt{4 + \sigma^2}}$$

Thus, since $\sum_k t_k^2 = \|\theta_i^* - \theta_j^*\|^2$:

$$\|c_i(\boldsymbol{\theta}) - c_j(\boldsymbol{\theta})\|^2 \geq \sum_k \left( \frac{|t_k|}{\sqrt{4 + \sigma^2}} \right)^2 \geq \frac{\|\theta_i^* - \theta_j^*\|^2}{4 + \sigma^2}$$

However, since $\forall i \neq j$, $\|\theta_i^* - \theta_j^*\| \geq \Delta$, it follows:

$$\|c_i(\boldsymbol{\theta}) - c_j(\boldsymbol{\theta})\|^2 \geq \frac{\Delta^2}{4 + \sigma^2}$$

since, by assumption, $\sigma \ll 1$. Thus, for $c \approx 2$,

$$\forall i \neq j, \|c_i(\boldsymbol{\theta}) - c_j(\boldsymbol{\theta})\| \geq \frac{\Delta}{c}. \tag{1}$$

It is easy to see that the proof holds for any set $\boldsymbol{\theta}$ of ortho-normal unit vectors. Specifically, since vectors drawn randomly from a $d$-dimensional unit sphere are nearly orthogonal, for large $d$, the result holds for them as well. $\qquad\square$

We now prove an analogue of the Lemma A.1 for the $K$ models of $\boldsymbol{\theta}$ that satisfy the proximity condition.

**Lemma A.2.** *Let $\boldsymbol{\theta}$ be collection of $K$ models that satisfy the $\Delta$-proximity condition. Then, for any $i \neq j$, $\|c_i(\boldsymbol{\theta}) - c_j(\boldsymbol{\theta})\| > \frac{\Delta}{2}$.*

*Proof.* Recall, $c_k(\boldsymbol{\theta}) = [c_k^1, c_k^2, \ldots c_k^N]$, where $c_k^j = \mathbb{E}\left[F_k^i(\theta_j)\right] = \alpha(\theta_j, \theta_k^*) = \sqrt{\|\theta - \theta_k^*\|^2 + \sigma^2}$.

Since $\|\theta_i - \theta_i^*\| \leq \frac{\Delta}{4}$ and since $\|\theta_i^* - \theta_j^*\| \geq \Delta$, by triangle inequality it follows that $\|\theta_i - \theta_j^*\| \geq \frac{3\Delta}{4}$. Likewise, $\|\theta_j - \theta_j^*\| \leq \frac{\Delta}{4}$ and $\|\theta_j - \theta_i^*\| \geq \frac{3\Delta}{4}$. Thus, since $\sigma \ll \frac{\Delta}{4}$:

$$|c_j^i - c_i^i| = \left| \sqrt{\|\theta_i - \theta_j^*\|^2 + \sigma^2} - \sqrt{\|\theta_i - \theta_i^*\|^2 + \sigma^2} \right|$$

$$\geq \left| \|\theta_i - \theta_j^*\|^2 - \sqrt{\|\theta_i - \theta_i^*\|^2 + \left(\frac{\Delta}{4}\right)^2} \right| \geq \frac{3 - \sqrt{2}}{4}\Delta$$

Likewise,

$$|c_i^j - c_j^j| = \left| \sqrt{\|\theta_j - \theta_i^*\|^2 + \sigma^2} - \sqrt{\|\theta_j - \theta_j^*\|^2 + \sigma^2} \right| \geq \frac{3 - \sqrt{2}}{4}\Delta$$

Since:

$$\|c_i(\boldsymbol{\theta}) - c_j(\boldsymbol{\theta})\|^2 \geq |c_j^i - c_i^i|^2 + |c_i^j - c_j^j|^2,$$

it follows:

$$\|c_i(\boldsymbol{\theta}) - c_j(\boldsymbol{\theta})\|^2 \geq 2\left(\frac{3 - \sqrt{2}}{4}\right)^2 \Delta^2.$$

Thus,

$$\|c_i(\boldsymbol{\theta}) - c_j(\boldsymbol{\theta})\| > \frac{\Delta}{2}$$

$\qquad\square$

Any ortho-normal model, by definition, satisfies $\|\theta\| = 1 < 2$. Now we show the same for any $\Delta$-proximal model.

**Lemma A.3.** $\|\theta\| < 2$, *for any $\Delta$-proximal model $\theta$.*

*Proof.* This follows from our assumption $\|\theta_k^*\| = 1$ for all $k \in [K]$ and since there exists a $k \in [K]$ such that $\|\theta - \theta_k^*\| \leq \Delta/4$, where $\Delta < 2$. Therefore, by triangle inequality, $\|\theta\| < 2$. $\qquad\square$

Since, for all models, $\|\theta\| < 2$, this implies $\|\theta - \theta_k^*\| \leq 3$, for all $k \in [K]$. Thus, it follows, for all $k \in [K]$:

$$\frac{1}{2n_k} \left( \|\theta - \theta_k^*\|^2 + \sigma^2 \right) \leq \frac{1}{2n_k} \left( 9 + \sigma^2 \right) \tag{2}$$

All our results below hold for any ortho-normal or $\Delta$-proximal models collection $\boldsymbol{\theta}$, unless noted otherwise.

The Lemma below establishes that SLVs of a cluster are concentrated around their means and this result holds with high probability.

**Lemma A.4.** *Let client $i$ be in cluster $s(i)$. Let $Var_j(a_{s(i)}^i)$ be the variance of the $j$-th entry of its SLV $a_{s(i)}^i(\boldsymbol{\theta})$. Then, for $t = \frac{9}{2} \log^2 \frac{NM}{\delta}$ and for any error tolerance $\delta < 1$:*

$$\mathbb{P}\left[ \forall i, \ \|a_{s(i)}^i(\boldsymbol{\theta}) - c_{s(i)}(\boldsymbol{\theta})\|^2 \leq t \max_j Var_j(a_{s(i)}^i) \right] \geq 1 - \delta.$$

*Proof.* In what follows, we use $k$ to denote the cluster assignment $s(i)$ for client $i$. Let

$$a_k^i(\boldsymbol{\theta}) = \left[ F_k^i(\theta_1), F_k^i(\theta_2), \ldots F_k^i(\theta_N) \right],$$

and therefore

$$c_k(\boldsymbol{\theta}) = \left[ \mathbb{E}[F_k^i(\theta_1)], \mathbb{E}[F_k^i(\theta_2)], \ldots, \mathbb{E}[F_k^i(\theta_N)] \right].$$

We start by focusing on the $j$-th component $F_k^i(\theta_j)$ of $a_k^i(\boldsymbol{\theta})$. For notational convenience, we will omit the index $j$ and refer to this component as $F_k^i(\theta)$ in the following proof.

Recall

$$\mathcal{L}_k^i(\theta) = \left( F_k^i(\theta) \right)^2 = \frac{1}{n_k} \sum_{j=1}^{n_k} \left( \langle \theta_k^* - \theta, x_j \rangle + \epsilon_j \right)^2.$$

$\mathcal{L}_k^i(\theta)$ is scaled chi-squared random variable with $n_k$ degrees of freedom (i.e. $\sim \frac{\alpha(\theta, \theta_k^*)^2}{n_k} \chi^2(n_k)$). Thus, its mean and variance are:

$$\mathbb{E}\left[ \mathcal{L}_k^i(\theta) \right] = \alpha(\theta, \theta_k^*)^2$$

$$\mathrm{Var}(\mathcal{L}_k^i(\theta)) = \frac{2}{n_k} \alpha(\theta, \theta_k^*)^4$$

Note

$$\mathcal{L}_k^i(\theta) \sim \sum_j \frac{1}{n_k} \alpha(\theta, \theta_k^*)^2 y_j^2$$

where $y_j \sim \mathcal{N}(0, 1)$. Thus,

$$\mathcal{L}_k^i(\theta) \sim \sum_j \frac{\mathbb{E}\left[ \mathcal{L}_k^i(\theta) \right]}{n_k} y_j^2$$

Let $\gamma = \sqrt{\sum_j \left( \frac{\mathbb{E}[\mathcal{L}_k^i(\theta)]}{n_k} \right)^2} = \frac{1}{\sqrt{n_k}} \mathbb{E}\left[ \mathcal{L}_k^i(\theta) \right]$. Let $\alpha = \frac{\mathbb{E}[\mathcal{L}_k^i(\theta)]}{n_k}$. We apply Lemma 1, Section 4 of Laurent & Massart (2000) to get:

$$\mathbb{P}\left[ \mathcal{L}_k^i(\theta) - \mathbb{E}\left[ \mathcal{L}_k^i(\theta) \right] \geq 2\alpha\sqrt{x} + 2\gamma x \right] \leq e^{-x}$$

Here

$$2\alpha\sqrt{x} + 2\gamma x = \mathbb{E}\left[ \mathcal{L}_k^i(\theta) \right] \left( \frac{2\sqrt{x}}{n_k} + \frac{2x}{\sqrt{n_k}} \right)$$

Note that $\frac{3x}{\sqrt{n_k}} > \frac{2\sqrt{x}}{n_k} + \frac{2x}{\sqrt{n_k}}$ for $x > 1$ (specifically for any $x > \frac{4}{n_k}$). Thus:

$$\mathbb{P}\left[ \mathcal{L}_k^i(\theta) - \mathbb{E}\left[ \mathcal{L}_k^i(\theta) \right] \geq \mathbb{E}\left[ \mathcal{L}_k^i(\theta) \right] \frac{3x}{\sqrt{n_k}} \right] \leq e^{-x}$$

With $3x = 2(1 + \beta)$:

$$\mathbb{P}\left[\mathcal{L}_k^i(\theta) \geq \mathbb{E}\left[\mathcal{L}_k^i(\theta)\right]\left(1 + 2(1+\beta)\frac{1}{\sqrt{n_k}}\right)\right] \leq e^{-\frac{2}{3}(1+\beta)}$$

Recall $\mathcal{L}_k^i(\theta) = \left(F_k^i(\theta)\right)^2$ and

$$\mathbb{E}\left[\mathcal{L}_k^i(\theta)\right] = \alpha(\theta, \theta_k^*)^2 = \mathbb{E}\left[F_k^i(\theta)\right]^2$$

Thus:

$$\mathbb{P}\left[F_k^i(\theta)^2 \geq \mathbb{E}\left[F_k^i(\theta)\right]^2\left(1 + 2(1+\beta)\frac{1}{\sqrt{n_k}}\right)\right] \leq e^{-\frac{2}{3}(1+\beta)}$$

Since

$$1 + 2(1+\beta)\frac{1}{\sqrt{n_k}} \leq \left(1 + (1+\beta)\frac{1}{\sqrt{n_k}}\right)^2,$$

$$\mathbb{P}\left[F_k^i(\theta) \geq \mathbb{E}\left[F_k^i(\theta)\right]\left(1 + (1+\beta)\frac{1}{\sqrt{n_k}}\right)\right] \leq e^{-\frac{2}{3}(1+\beta)}$$

or

$$\mathbb{P}\left[\left(F_k^i(\theta) - \mathbb{E}\left[F_k^i(\theta)\right]\right)^2 \geq (1+\beta)^2\frac{\mathbb{E}\left[F_k^i(\theta)\right]^2}{n_k}\right] \leq e^{-\frac{2}{3}(1+\beta)}$$

Since

$$\mathrm{Var}\left(F_k^i(\theta)\right) = \frac{\alpha(\theta, \theta_k^*)^2}{2n_k} = \frac{\mathbb{E}\left[F_k^i(\theta)\right]^2}{2n_k},$$

$$\mathbb{P}\left[\left(F_k^i(\theta) - \mathbb{E}\left[F_k^i(\theta)\right]\right)^2 \geq 2(1+\beta)^2\mathrm{Var}\left(F_k^i(\theta)\right)\right] \leq e^{-\frac{2}{3}(1+\beta)}$$

We set $t = 2(1 + \beta)^2$. Then:

$$\mathbb{P}\left[\left(F_k^i(\theta) - \mathbb{E}\left[F_k^i(\theta)\right]\right)^2 \geq t\,\mathrm{Var}\left(F_k^i(\theta)\right)\right] \leq e^{-\frac{\sqrt{2t}}{3}}$$

Note that this result holds for a single component $F_k^i(\theta_j)$ of $A_i(\boldsymbol{\theta})$. By using union bound over all of its $N$ components:

$$\mathbb{P}\left[\sum_{j=1}^N \left(F_k^i(\theta_j) - \mathbb{E}\left[F_k^i(\theta_j)\right]\right)^2 \geq t\max_j \mathrm{Var}\left(F_k^i(\theta_j)\right)\right] \leq Ne^{-\frac{\sqrt{2t}}{3}}$$

or

$$\mathbb{P}\left[\left\|a_k^i(\boldsymbol{\theta}) - c_k(\boldsymbol{\theta})\right\|^2 \geq t\max_j \mathrm{Var}\left(F_k^i(\theta_j)\right)\right] \leq Ne^{-\frac{\sqrt{2t}}{3}}.$$

Since $k = s(i)$ and $\mathrm{Var}(F_k^i(\theta_j)) = \mathrm{Var}_j(a_{s(i)}^i)$:

$$\mathbb{P}\left[\left\|a_{s(i)}^i(\boldsymbol{\theta}) - c_{s(i)}(\boldsymbol{\theta})\right\|^2 \geq t\max_j \mathrm{Var}_j(a_{s(i)}^i)\right] \leq Ne^{-\frac{\sqrt{2t}}{3}}.$$

Applying union bounds for all $M$ clients and by noticing that for $t = \frac{9}{2}\log^2\frac{NM}{\delta}$:

$$MNe^{-\frac{\sqrt{2t}}{3}} = \delta$$

we get

$$\mathbb{P}\left[\forall i, \left\|a_{s(i)}^i(\boldsymbol{\theta}) - c_{s(i)}(\boldsymbol{\theta})\right\|^2 \geq t\max_j \mathrm{Var}_j(a_{s(i)}^i)\right] \leq \delta$$

or

$$\mathbb{P}\left[\forall i, \left\|a_{s(i)}^i(\boldsymbol{\theta}) - c_{s(i)}(\boldsymbol{\theta})\right\|^2 \leq t\max_j \mathrm{Var}_j(a_{s(i)}^i)\right] \geq 1 - \delta.$$

$\square$

For models $\boldsymbol{\theta}$, Let $A(\boldsymbol{\theta})$ be the $M \times N$ matrix of the SLVs of all $M$ clients and let $C(\boldsymbol{\theta})$ be the corresponding $M \times N$ matrix of SLV means. Thus, $X(\boldsymbol{\theta}) = A(\boldsymbol{\theta}) - C(\boldsymbol{\theta})$ is the matrix of centered SLVs for all clients. Let $X_i(\boldsymbol{\theta}) = A_i(\boldsymbol{\theta}) - C_i(\boldsymbol{\theta})$ be the $i$-th row of $A(\boldsymbol{\theta}) - C(\boldsymbol{\theta})$, with $A_i(\boldsymbol{\theta})$ being the SLV of the $i$-th client and $C_i(\boldsymbol{\theta})$ the mean of that SLV.

We now establish a bound on the directional variance of the rows of $A(\boldsymbol{\theta}) - C(\boldsymbol{\theta})$. First we show the following:

**Lemma A.5.** *Let $X$ be an $N$-dimensional vector with components $x_i$ such that $\mathbb{E}[x_i] = 0$ for all $i$, and let $v$ be any $N$-dimensional vector. Then*

$$\mathbb{E}\left[\langle X, v \rangle^2\right] \leq \left(\sum_{i=1}^{N} |v_i| \sqrt{Var(x_i)}\right)^2. \tag{3}$$

*Proof.* Note

$$\mathbb{E}\left[\langle X, v \rangle^2\right] = \sum_{i=1}^{N} \sum_{j=1}^{N} v_i v_j \mathbb{E}[x_i x_j] \leq \sum_{i=1}^{N} \sum_{j=1}^{N} |v_i v_j \mathbb{E}[x_i x_j]|.$$

Splitting the sum into diagonal and off-diagonal terms:

$$\mathbb{E}\left[\langle X, v \rangle^2\right] \leq \sum_{i=1}^{N} v_i^2 \mathbb{E}[x_i^2] + \sum_{i \neq j} |v_i v_j \mathbb{E}[x_i x_j]|.$$

Applying the Cauchy-Schwarz inequality to the off-diagonal terms:

$$|\mathbb{E}[x_i x_j]| \leq \sqrt{\mathbb{E}[x_i^2]\mathbb{E}[x_j^2]} \quad \text{for} \quad i \neq j.$$

Thus,

$$\mathbb{E}\left[\langle X, v \rangle^2\right] \leq \sum_{i=1}^{N} v_i^2 \mathbb{E}[x_i^2] + \sum_{i \neq j} |v_i v_j| \sqrt{\mathbb{E}[x_i^2]\mathbb{E}[x_j^2]}.$$

Thus,

$$\mathbb{E}\left[\langle X, v \rangle^2\right] \leq \left(\sum_{i=1}^{N} |v_i| \sqrt{\mathbb{E}[x_i^2]}\right)^2.$$

We substitute $\mathbb{E}\left[x_i^2\right] = \mathrm{Var}(x_i)$ (since $\mathbb{E}[x_i] = 0$) to express the bound in terms of variances:

$$\mathbb{E}\left[\langle X, v \rangle^2\right] \leq \left(\sum_{i=1}^{N} |v_i| \sqrt{\mathrm{Var}(x_i)}\right)^2.$$

$\square$

**Lemma A.6.** *Let $X_i(\boldsymbol{\theta}) = A_i(\boldsymbol{\theta}) - C_i(\boldsymbol{\theta})$ be the $i$-th row of $A(\boldsymbol{\theta}) - C(\boldsymbol{\theta})$, with $A_i(\boldsymbol{\theta})$ being the SLV of $i$-th client and $C_i(\boldsymbol{\theta})$ the mean of that SLV. Let $i$-th client be in in cluster $k$.*

$$\max_{v:\|v\|=1} \mathbb{E}\left[\langle X_i(\boldsymbol{\theta}), v \rangle^2\right] \leq N \frac{1}{2n_k}\left(9 + \sigma^2\right) \tag{4}$$

*Proof.* Applying Equation 3 of Lemma A.5, it follows:

$$\max_{v:\|v\|=1} \mathbb{E}\left[\langle X_i(\boldsymbol{\theta}), v \rangle^2\right] \leq \max_{v:\|v\|=1} \left(\sum_{j=1}^{N} |v_j| \sqrt{\mathrm{Var}(X_i(\theta_j))}\right)^2,$$

where $X_i(\theta_j)$ is the $j$-th component of $X_i(\boldsymbol{\theta})$, which is the centered square root loss of client $i$ on model $\theta_j$. That is $X_i(\theta_j) = A_i(\theta_j) - \mathbb{E}[A_i(\theta_j)]$. Therefore, $\mathrm{Var}(X_i(\theta_j)) = \mathrm{Var}(A_i(\theta_j))$. Hence:

$$\max_{v:\|v\|=1} \mathbb{E}\left[\langle X_i(\boldsymbol{\theta}), v \rangle^2\right] \leq \max_{v:\|v\|=1} \left(\sum_{j=1}^{N} |v_j| \sqrt{\mathrm{Var}(A_i(\theta_j))}\right)^2$$

Combining with Equation 2:

$$\text{Var}(A_i(\theta_j)) = \frac{1}{2n_k}\left(\|\theta_j - \theta_k^*\|^2 + \sigma^2\right) \leq \frac{1}{2n_k}\left(9 + \sigma^2\right).$$

Thus

$$\max_{v:\|v\|=1} \mathbb{E}\left[\langle X_i(\boldsymbol{\theta}), v\rangle^2\right] \leq N\frac{1}{2n_k}\left(9 + \sigma^2\right)$$

since maximum happens when all $v_j = \frac{1}{\sqrt{N}}$. $\qquad\square$

Using a similar argument, it follows:

$$\mathbb{E}\left[\|X_i(\boldsymbol{\theta})\|^2\right] \leq N\frac{1}{2n_k}\left(9 + \sigma^2\right) \qquad (5)$$

We now establish a bound on the matrix operator norm $\|X(\boldsymbol{\theta})\| = \|A(\boldsymbol{\theta}) - C(\boldsymbol{\theta})\|$. Let $n = \min_k n_k$ be the smallest number of data points any client has. Let $\gamma = \sqrt{N\frac{1}{2n}\left(9 + \sigma^2\right)}$. Then we can show the following bound:

**Lemma A.7.** *With probability at least* $1 - \frac{1}{polylog(M)}$:

$$\|X(\boldsymbol{\theta})\| \leq \gamma\sqrt{M}polylog(M) \qquad (6)$$

*Proof.* With abuse of notation, we will use $X_i$ to refer to $i$-th row of $\mathbf{X}(\boldsymbol{\theta})$ and use $X_{ij}$ to denote the $j$-th entry of this row. Since $\text{Var}(\|X_i\|) \leq E[\|X_i\|^2]$, then by applying Equation 5 and by definition of $\gamma$, it follows $\text{Var}(\|X_i\|) \leq E[\|X_i\|^2] \leq \gamma^2$. Thus, by Chebyshev's inequality:

$$\mathbb{P}\left[\|X_i\| \geq \gamma\sqrt{M}\text{polylog}(M)\right] \leq \frac{\gamma^2}{M\gamma^2\text{polylog}(M)}$$

Thus by union bound

$$\mathbb{P}\left[\max_{i=1,\ldots,M}\|X_i\| \geq \gamma\sqrt{M}\text{polylog}(M)\right] \leq \frac{1}{\text{polylog}(M)}$$

Thus with probability at least $1 - \frac{1}{\text{polylog}(M)}$:

$$\max_{i=1,\ldots,M}\|X_i\| < \gamma\sqrt{M}\text{polylog}(M)$$

Now for any unit column vector $v$

$$v^T E\left[\mathbf{X}(\boldsymbol{\theta})^T\mathbf{X}(\boldsymbol{\theta})\right]v = \sum_{i=1}^{M} E\left[v^T X_i^T X_i v\right] = \sum_{i=1}^{M} E\left[\|X_i v\|^2\right] = \sum_{i=1}^{M} E\left[\langle X_i, v\rangle^2\right]$$

Thus from Equation 4 of Lemma A.6 and by definition of $\gamma$, it follows:

$$\max_{v:\|v\|=1} v^T E\left[\mathbf{X}(\boldsymbol{\theta})^T\mathbf{X}(\boldsymbol{\theta})\right]v \leq \gamma^2 M$$

Since $E\left[\mathbf{X}(\boldsymbol{\theta})^T\mathbf{X}(\boldsymbol{\theta})\right]$ is symmetric, its spectral norm equals its largest eigenvalue, which is what is bounded above. Thus

$$\left\|E\left[\mathbf{X}(\boldsymbol{\theta})^T\mathbf{X}(\boldsymbol{\theta})\right]\right\| \leq \gamma^2 M$$

We now use the following fact (Fact 6.1) from Kumar & Kannan (2010) that follows from the result of Dasgupta et al. (2007).

**Lemma A.8.** *Let $Y$ be a $n \times d$ matrix, $n \geq d$, whose rows are chosen independently. Let $Y_i$ denote its $i$-th row. Let there exist $\gamma$ such that $\max_i \|Y_i\| \leq \gamma\sqrt{n}$ and $\|E\left[Y^TY\right]\| \leq \gamma^2 n$. Then $\|Y\| \leq \gamma\sqrt{n}polylog(n)$.*

Thus, it follows that with probability at least $1 - \dfrac{1}{\text{polylog}(M)}$:

$$\|\mathbf{X}(\boldsymbol{\theta})\| \leq \gamma\sqrt{M}\text{polylog}(M)$$

$\square$

We now establish one of our main results, that the SLVs satisfy the proximity condition of Kumar & Kannan (2010), thereby proving that k-means with suitably initialized centers recovers the accurate clustering of the SLVs (i.e. it recovers an accurate clustering of the clients).

Recall that for client $i$ of cluster $k$, its SLV is $a_k^i(\boldsymbol{\theta}) = \left[F_k^i(\theta_1), F_k^i(\theta_2), \dots F_k^i(\theta_N)\right]$. Also $c_k(\boldsymbol{\theta}) = \left[c_1^k, c_2^k, \dots c_N^k\right]$, is the mean of $k$-th clusters SLVs, where $c_k^j = \mathbb{E}\left[F_k^i(\theta_j)\right]$. Recall $X(\boldsymbol{\theta}) = A(\boldsymbol{\theta}) - C(\boldsymbol{\theta})$ is the matrix of centered SLVs for all clients. Also, $X_i(\boldsymbol{\theta}) = A_i(\boldsymbol{\theta}) - C_i(\boldsymbol{\theta})$ denotes the $i$-th row of $A(\boldsymbol{\theta}) - C(\boldsymbol{\theta})$. Let $m_k$ denote the number of clients in cluster $k$.

**Theorem A.9.** *When $nM$, which is a lower bound on the the number of data points across all clients, is much larger in comparison to $d, K$, and specifically when*

$$dK^4\, polylog(M) = o(nM), \tag{7}$$

*then for all ortho-normal or proximal models $\boldsymbol{\theta}$, the following proximity condition of Kumar & Kannan (2010) holds for the SLV of any client $i$, with probability at least $1 - \delta - \dfrac{1}{polylog(M)}$, for any error tolerance $\delta$ :*

$$\forall\, j' \neq j,\ \left\|a_j^i(\boldsymbol{\theta}) - c_{j'}(\boldsymbol{\theta})\right\| - \left\|a_j^i(\boldsymbol{\theta}) - c_j(\boldsymbol{\theta})\right\| \geq \left(\frac{c'K}{m_i} + \frac{c'K}{m_j}\right)\|A(\boldsymbol{\theta}) - C(\boldsymbol{\theta})\|$$

*for some large constant $c'$. Here it is assumed that client $i$ belongs to cluster $j$.*

*Proof.* Since $\text{Var}(A_i(\theta_j)) = \frac{1}{2n_k}\left(\|\theta_i - \theta_k^*\|^2 + \sigma^2\right)$, it follows that $\forall\, i, j,\ \text{Var}(A_i(\theta_j)) \leq \frac{1}{2n}\left(9 + \sigma^2\right) = \gamma^2/N$, where $n = \min_k n_k$. Combining this with Equation A.4 established in Lemma A.4, it follows with high probability (at least $1 - \delta$):

$$\forall\, i,\ \|A_i(\boldsymbol{\theta}) - C_i(\boldsymbol{\theta})\|^2 \leq \frac{9}{2}\log^2\frac{NM}{\delta}\max_j\text{Var}(A_i(\theta_j)) \leq \frac{9}{2}\log^2\frac{NM}{\delta}\frac{\gamma^2}{N},$$

where $A_i(\boldsymbol{\theta})$ and $C_i(\boldsymbol{\theta})$ are the $i$-th rows of matrices $A(\boldsymbol{\theta})$ and $C(\boldsymbol{\theta})$ respectively.

Without loss of generality, let the SLV of client $i$ belonging to cluster $j$ be in the $i$-th row of the matrix $A(\boldsymbol{\theta})$. Thus, it follows:

$$\left\|a_j^i(\boldsymbol{\theta}) - c_j(\boldsymbol{\theta})\right\| = \|A_i(\boldsymbol{\theta}) - C_i(\boldsymbol{\theta})\| \leq \frac{3}{\sqrt{2}}\frac{1}{\sqrt{N}}\log\frac{NM}{\delta}\gamma.$$

Let

$$\zeta = \frac{3}{\sqrt{2}}\frac{1}{\sqrt{N}}\log\frac{NM}{\delta}\gamma$$

Since models in $\boldsymbol{\theta}$ collection are ortho-normal or $\Delta$-proximal, from Lemma A.1 and Lemma A.2 it follows that for any $j' \neq j$, $\|c_{j'}(\boldsymbol{\theta}) - c_j(\boldsymbol{\theta})\| \geq \Delta/c$, where $c \approx 2$.

Thus, by triangle inequality, for any $j' \neq j$:

$$\left\|a_j^i(\boldsymbol{\theta}) - c_{j'}(\boldsymbol{\theta})\right\| - \left\|a_j^i(\boldsymbol{\theta}) - c_j(\boldsymbol{\theta})\right\| \geq \frac{\Delta}{c} - 2\zeta \tag{8}$$

We now show that:

$$\frac{\Delta}{c} - 2\zeta \geq \left(\frac{c'K}{m_i} + \frac{c'K}{m_j}\right)\gamma\sqrt{M}\text{polylog}(M)$$

By assumption, $N \leq d \ll M$. Thus, in

$$2\zeta + \left(\frac{c'K}{m_i} + \frac{c'K}{m_j}\right)\gamma\sqrt{M}\text{polylog}(M) = \frac{3}{\sqrt{2}}\frac{1}{\sqrt{N}}\log\frac{NM}{\delta}\gamma + \left(\frac{c'K}{m_i} + \frac{c'K}{m_j}\right)\gamma\sqrt{M}\text{polylog}(M),$$

the second term dominates. Also, we make the assumption that number of clients in each cluster is not too far from the average number of clients per cluster $M/K$. That is, there exists a small constant $c''$ such that the number of clients $m_k$ in any cluster $k$ is at least $\frac{1}{c''}\frac{M}{K}$. Hence:

$$2\zeta + \left(\frac{c'K}{m_i} + \frac{c'K}{m_j}\right)\gamma\sqrt{M}\text{polylog}(M) = O\left(\gamma\frac{K^2}{\sqrt{M}}\text{polylog}(M)\right)$$

Since $\sigma \ll 1$ is very small and $K \le N \le d$, it follows that $\gamma \le \sqrt{\frac{2d}{n}}$. Thus:

$$2\zeta + \left(\frac{c'K}{m_i} + \frac{c'K}{m_j}\right)\gamma\sqrt{M}\text{polylog}(M) = O\left(\frac{\sqrt{d}}{\sqrt{n}}\frac{K^2}{\sqrt{M}}\text{polylog}(M)\right)$$

Since by assumption $\Delta$ is close to 1 and

$$dK^4\,\text{polylog}(M) = o(nM),$$

it follows that $\frac{\Delta}{c}$ (where $c \approx 2$) is much larger than $O\left(\frac{\sqrt{d}}{\sqrt{n}}\frac{K^2}{\sqrt{M}}\text{polylog}(M)\right)$.

This establishes, with probability at least $1 - \delta$:

$$\frac{\Delta}{c} \ge 2\zeta + \left(\frac{c'K}{m_i} + \frac{c'K}{m_j}\right)\gamma\sqrt{M}\text{polylog}(M)$$

Applying Equation 6 of Lemma A.7, it follows with probability at least $1 - \delta - \dfrac{1}{\text{polylog}(M)}$ that:

$$\frac{\Delta}{c} - 2\zeta \ge \left(\frac{c'K}{m_i} + \frac{c'K}{m_j}\right)\|A(\boldsymbol{\theta}) - C(\boldsymbol{\theta})\|$$

Combining with Equation 8, the proximity result follows. That is, with probability at least $1 - \delta - \dfrac{1}{\text{polylog}(M)}$:

$$\forall\, j' \ne j,\; \left\|a_j^i(\boldsymbol{\theta}) - c_{j'}(\boldsymbol{\theta})\right\| - \left\|a_j^i(\boldsymbol{\theta}) - c_j(\boldsymbol{\theta})\right\| \ge \left(\frac{c'K}{m_i} + \frac{c'K}{m_j}\right)\|A(\boldsymbol{\theta}) - C(\boldsymbol{\theta})\|$$

$\square$

This yields one of our main results:

**Theorem A.10.** *For ortho-normal or $\Delta$-proximal models collection $\boldsymbol{\theta}$, with probability at least $1 - \delta - \dfrac{1}{\text{polylog}(M)}$, k-means with suitably initialized centers recovers the accurate clustering of the SLVs (and hence an accurate clustering of the clients) in one shot.*

*Proof.* Follows from Lemma A.1 and Theorem A.9 and by applying the result of Kumar & Kannan (2010). $\square$

*Remark* A.11. There have been further improvements to the proximity bounds of Kumar & Kannan (2010) (e.g. Awasthi & Sheffet (2012)) that can be used to improve our bounds (Equation 7). However, since our goal in this work is mainly to establish feasibility, we leave any improvements of those bounds for future work.

*Remark* A.12. In the proof above we used loss vectors of square roots of losses (SLV). This is because it allows us to recover the accurate clustering of SLVs in one round of k-means. For loss vectors (LV) formed by losses instead of square roots of losses, we are only able to prove that the distance bound as established in Lemma A.4 holds for a large fraction of the rows $A(\boldsymbol{\theta})$, which means that k-means can recover accurate clustering of not all but still a large number of clients. This is still fine for our approach, since we do not need to apply updates from all clients to build the models for the next round. However, it does mean that the analysis gets more complex and we therefore leave that for future work.

In the following, let the set of clients in cluster $k$ be denoted by $Q_k$. Let $i \in Q_k$ be a client in cluster $k$. Let $H_k^i$ denote the set of $n_k$ data points and additive noise values $(x_j, y_j, \epsilon_j)$ of client $i$. Its empirical loss is

$$\mathcal{L}_k^i \left( \theta; H_k^i \right) = \frac{1}{n_k} \sum_{(x_j, y_j, \epsilon_j) \in H_k^i} \left( \langle \theta, x_j \rangle - y_j \right)^2 = \frac{1}{n_k} \sum_{(x_j, y_j, \epsilon_j) \in H_k^i} \left( \langle \theta_k^* - \theta, x_j \rangle + \epsilon_j \right)^2$$

Let $H_k$ be the $m_k n_k \times d$ matrix with rows being the features of clients in cluster $k$:

$$H_k = \left[ x_j : (x_j, y_j, \epsilon_j) \in \bigcup_{j \in Q_k} H_k^j \right]$$

Note that all rows of $H_k$ are drawn independently from $\mathcal{N}(0, I_d)$. Let $\xi_k$ be the $m_k n_k \times 1$ column vector of additive noise parameters of clients in cluster $k$:

$$\xi_k = \left[ \epsilon_j : (x_j, y_j, \epsilon_j) \in \bigcup_{j \in Q_k} H_k^j \right]$$

Note that all entries of $\xi_k$ are drawn independently from $\mathcal{N}(0, \sigma^2)$.

**Lemma A.13.** *After applying gradient updates of clients belonging to a cluster $k$ to a model $\theta, \|\theta\| \leq 2$, the new model $\theta_k$ satisfies $\|\theta_k^* - \theta_k\| \leq \Delta/c$ with probability at least $1 - c_6 e^{-c_7 d}$, for some constants $c_6, c_7$. Furthermore, with probability at least $1 - c_6 K e^{-c_7 d}$, $\|\theta_k^* - \theta_k\| \leq \frac{\Delta}{c}$, for all clusters $k$. Here learning parameter $\eta$ is assumed to be at most $\frac{1}{2}\left(1 - \frac{\Delta}{3c}\right)$.*

*Proof.* Note that

$$\nabla \mathcal{L}_k^i \left( \theta; H_k^i \right) = \frac{1}{n_k} \sum_{(x_j, y_j) \in H_k^i} \left( 2 x_j x_j^T (\theta - \theta_k^*) - 2 x_j \epsilon_j \right)$$

Gradient updates from clients $Q_k$ of cluster $k$ on model $\theta$ for learning rate $\eta$ yield model $\theta_k$:

$$\theta_k = \theta - \frac{\eta}{m_k} \sum_{i \in Q_k} \nabla \mathcal{L}_k^i(\theta; H_k^i)$$

Thus, as rows of matrix $H_k$ contain features of all clients in cluster $k$, and since entries of vector $\xi_k$ contain corresponding additive noise parameters of those clients:

$$\theta_k = \theta - \frac{\eta}{m_k n_k} \left( 2 H_k^T H_k (\theta - \theta_k^*) - 2 H_k^T \xi_k \right)$$

Thus,

$$\theta_k - \theta_k^* = \theta - \theta_k^* - \frac{2\eta}{m_k n_k} H_k^T H_k (\theta - \theta_k^*) + \frac{2\eta}{m_k n_k} H_k^T \xi_k$$

Or

$$\theta_k - \theta_k^* = \theta - \theta_k^* - \frac{2\eta}{m_k n_k} \mathbb{E}[H_k^T H_k](\theta - \theta_k^*) + \frac{2\eta}{m_k n_k} \left( \mathbb{E}[H_k^T H_k] - H_k^T H_k \right)(\theta - \theta_k^*) + \frac{2\eta}{m_k n_k} H_k^T \xi_k$$

Note $\mathbb{E}\left[ H_k^T H_k \right] = m_k n_k I$. Therefore:

$$\theta_k - \theta_k^* = (1 - 2\eta)(\theta - \theta_k^*) + \frac{2\eta}{m_k n_k} \left( \mathbb{E}\left[ H_k^T H_k \right] - H_k^T H_k \right)(\theta - \theta_k^*) + \frac{2\eta}{m_k n_k} H_k^T \xi_k$$

Taking norms:

$$\|\theta_k - \theta_k^*\| \leq |1 - 2\eta| \|\theta - \theta_k^*\| + \frac{2\eta}{m_k n_k} \left\| \mathbb{E}\left[ H_k^T H_k \right] - H_k^T H_k \right\| \|\theta - \theta_k^*\| + \frac{2\eta}{m_k n_k} \left\| H_k^T \xi_k \right\|$$

We now bound the various terms on the RHS.

Since $\|\theta\| \leq 2$ and $\|\theta_k^*\| \leq 1$:

$$\|\theta - \theta_k^*\| \leq 3.$$

As shown in Ghosh et al. (2022), following the result of Wainwright (2019), with probability at least $1 - 2e^{-\frac{d}{2}}$, the operator norm of the matrix $\mathbb{E}\left[H_k^T H_k\right] - H_k^T H_k$ is bounded from above:

$$\left\|\mathbb{E}\left[H_k^T H_k\right] - H_k^T H_k\right\| \leq 6\sqrt{dm_k n_k}.$$

By assumption $d \ll m_k n_k$. Therefore,

$$\left\|\mathbb{E}\left[H_k^T H_k\right] - H_k^T H_k\right\| = o(m_k n_k).$$

and therefore $\frac{2\eta}{m_k n_k}\left\|\mathbb{E}[H_k^T H_k] - H_k^T H_k\right\|\|\theta - \theta_k^*\| \leq c_2\Delta$, for some $c_2 \ll 1$.

As shown in Ghosh et al. (2022), with probability at least $1 - c_4 e^{-c_5 d}$, for some constants $c_4, c_5$.

$$\left\|H^T\xi_k\right\| \leq c_1\sigma\sqrt{dm_k n_k} = o(\sigma m_k n_k),$$

for some constant $c_1$. Since $\sigma \ll 1$, $\frac{2\eta}{m_k n_k}\left\|H_k^T\xi_k\right\| \leq c_3\Delta$, for some $c_3 \ll 1$.

Combining all the bounds, with probability at least $1 - c_6 e^{-c_7 d}$, for some constants $c_6, c_7$:

$$\|\theta_k - \theta_k^*\| \leq 3|1 - 2\eta| + (c_2 + c_3)\Delta.$$

For $\eta \leq \frac{1}{2}\left(1 - \frac{\Delta}{3c}\right)$, $3|1 - 2\eta| \leq \frac{\Delta}{c}$. Thus, with appropriate choice of $\eta$, with probability at least $1 - c_6 e^{-c_7 d}$:

$$\|\theta_k - \theta_k^*\| \leq \frac{\Delta}{c} + (c_2 + c_3)\Delta < \frac{\Delta}{c}.$$

Thus, by applying union bound, it follows that with probability at least $1 - c_6 K e^{-c_7 d}$, $\|\theta_k^* - \theta_k\| \leq \frac{\Delta}{c}$, for all clusters $k$. $\qquad\square$

We set $c = 4$ and $\eta = \frac{1}{2}\left(1 - \frac{\Delta}{12}\right)$. Thus, $\|\theta_k^* - \theta_k\| \leq \frac{\Delta}{4}$ for all clusters $k$ with probability at least $1 - c_6 K e^{-c_7 d}$.

**Corollary A.14.** *After applying the gradient updates as in Lemma A.13, with probability at least $1 - c_6 K e^{-c_7 d}$, $\|\theta_k - \theta_k^*\|$ is a constant fraction smaller than $\|\theta - \theta_k^*\|$. The result applies to all models $\theta$ whose norm is less than 2.*

*Proof.* Note that Equation A.2 can be restated as:

$$\|\theta_k - \theta_k^*\| \leq |1 - 2\eta|\|\theta - \theta_k^*\| + (c_2 + c_3)\Delta.$$

For our choice of $\eta$,

$$\|\theta_k - \theta_k^*\| \leq \frac{\Delta}{3c}\|\theta - \theta_k^*\| + (c_2 + c_3)\Delta.$$

Since $\Delta$ is close to 1 and since $c \geq 4$, and since $(c_2 + c_3)\Delta$ is negligibly small, it follows that $\|\theta_k - \theta_k^*\|$ is a constant fraction smaller than $\|\theta - \theta_k^*\|$, as long as $\|\theta - \theta_k^*\|$ is much larger than $(c_2 + c_3)\Delta$.

$\qquad\square$

In the following, $s(i) \in [1, K]$ denotes the ground-truth cluster assignment for client $i$. Let $Q$ be a clustering of the clients. That is, $Q = [Q_1, Q_2, \ldots, Q_K]$, where $Q_k$ is the set of clients in the $k$-th cluster of $Q$. Let $G(Q; \boldsymbol{\theta}) = (U \cup V, E)$ be a fully-connected bi-partite graph. Here, $U$ and $V$ are sets of nodes of size $K$ each, where $k$-th node of $U$ (or $V$) corresponds to the $k$-th cluster of $Q$. $E$ is the set of edges, with weight $w(k, j)$ of edge $(k, j)$ set to the total loss of clients in $Q_k$ on model $\theta_j$ of $\boldsymbol{\theta}$. That is:

$$w(k, j) = \sum_{i \in Q_k} \mathcal{L}_{s(i)}^i(\theta; H_{s(i)}^i).$$

**Assumption A.15.** $n_k m_k (\frac{\rho}{\rho+1})^2 \gg$ polylog($K$), for $\rho = \frac{\Delta^2}{\sigma^2}$. In the following, $\boldsymbol{\theta} = [\theta_1 \quad \theta_2 \quad \cdots \quad \theta_K]$ refers to a collection of $K$ models that satisfy the $\Delta$-proximity condition $\forall k \in [K], \|\theta_k - \theta_k^*\| \leq \frac{\Delta}{c}$, for $c \geq 4$.

**Lemma A.16.** *Let $\boldsymbol{\theta}$ be a collection of $K$ models that satisfy the $\Delta$-proximity condition. Let $Q$ be an accurate clustering of clients to clusters. Then, the probability that min-cost perfect matching $M$ of $G(Q; \boldsymbol{\theta})$ is $M = \{(k, k) : Q_k \in Q\}$ is at least $1 - c_1 e^{-c_2 \frac{1}{\log K^2} n_k m_k (\frac{\rho}{\rho+1})^2}$.*

*Proof.* Let clients of a cluster $Q_k$ achieve lower loss on a model $\theta_j$ than on model $\theta_k$, for $j \neq k$. Note that the probability of this is:

$$\mathbb{P}\left[\sum_{i \in Q_k} \mathcal{L}_{s(i)}^i\left(\theta_j; H_{s(i)}^i\right) < \sum_{i \in Q_k} \mathcal{L}_{s(i)}^i\left(\theta_k; H_{s(i)}^i\right)\right]$$

Since $\sum_{i \in Q_k} \mathcal{L}_{s(i)}^i(\theta; H_{s(i)}^i)$ is scaled chi-squared random variable with $n_k m_k$ degrees of freedom (i.e. $\sim \frac{m_k}{n_k}(\|\theta - \theta_k^*\|^2 + \sigma^2)\chi^2(n_k m_k)$), this probability is:

$$\mathbb{P}\left[\|\theta_j - \theta_k^*\|^2 + \sigma^2)\chi^2(n_k m_k) < (\|\theta_k - \theta_k^*\|^2 + \sigma^2)\chi^2(n_k m_k)\right]$$

Since $\|\theta_k - \theta_k^*\| \leq \frac{\Delta}{4} = \frac{\Delta}{2} - \frac{\Delta}{4}$ and since $\|\theta_k^* - \theta_j^*\| \geq \Delta$, by triangle inequality it follows that $\|\theta_k - \theta_j^*\| \geq \frac{3\Delta}{4} = \frac{\Delta}{2} - \frac{\Delta}{4}$. Thus, from the result of Ghosh et al. (2022), which is obtained by applying the concentration properties of chi-squared random variables (Wainwright, 2019), it follows that for some constants $c_1$ and $c_2$:

$$\mathbb{P}\left[(\|\theta_j - \theta_k^*\|^2 + \sigma^2)\chi^2(n_k m_k) < (\|\theta_k - \theta_k^*\|^2 + \sigma^2)\chi^2(n_k m_k)\right] \leq c_1 e^{-c_2 n_k m_k (\frac{\rho}{\rho+1})^2}$$

Hence:

$$\mathbb{P}\left[\sum_{i \in Q_k} \mathcal{L}_{s(i)}^i\left(\theta_j; H_{s(i)}^i\right) \leq \sum_{i \in Q_k} \mathcal{L}_{s(i)}^i\left(\theta_k; H_{s(i)}^i\right)\right] \leq c_1 e^{-c_2 n_k m_k \left(\frac{\rho}{\rho+1}\right)^2}$$

Thus, by union bound it follows:

$$\mathbb{P}\left[\exists \, k \neq j \text{ s.t. } \sum_{i \in Q_k} \mathcal{L}_{s(i)}^i\left(\theta_j; H_{s(i)}^i\right) \leq \sum_{i \in Q_k} \mathcal{L}_{s(i)}^i\left(\theta_k; H_{s(i)}^i\right)\right] \leq K^2 c_1 e^{-c_2 n_k m_k \left(\frac{\rho}{\rho+1}\right)^2}$$

Thus:

$$\mathbb{P}\left[\forall \, k \neq j, \sum_{i \in Q_k} \mathcal{L}_{s(i)}^i\left(\theta_k; H_{s(i)}^i\right) < \sum_{i \in Q_k} \mathcal{L}_{s(i)}^i\left(\theta_j; H_{s(i)}^i\right)\right]$$
$$\geq 1 - K^2 c_1 e^{-c_2 n_k m_k \left(\frac{\rho}{\rho+1}\right)^2}$$
$$\geq 1 - c_1 e^{-c_2 \frac{1}{\log K^2} n_k m_k \left(\frac{\rho}{\rho+1}\right)^2}$$

Thus, with probability close to 1, clients in cluster $Q_k$, achieve lowest loss on model $\theta_k$, for all $k$.

$\square$

### A.3 COMPARISON WITH *IFCA*

We demonstrate that the cluster recovery condition of *IFCA* implies loss vector separation for *CLoVE*. We assume a gap of at least $\delta > 0$ between the loss a client achieves on its best model and its second best model. The following Lemma shows that vectors of clients belonging to different clusters are at least $\sqrt{2}\delta$ apart.

**Lemma A.17.** *Let clients $i$ and $j$ achieve lowest loss on models $k_1$ and $k_2$ respectively ($k_1 \neq k_2$). Let there be at least a $\delta$ gap between the lowest loss and the second lowest loss achieved by a client on any model. Then $\|L_i - L_j\| \geq \sqrt{2}\delta$, where $L_i$ and $L_j$ are the loss vectors for clients $i$ and $j$.*

*Proof.* Let $l(x, y)$ denote loss of client $x$ on model $y$. Then:

$$\|L_i - L_j\|^2 \geq (l(i, k_1) - l(j, k_1))^2 + (l(i, k_2) - l(j, k_2))^2$$

However, because of $\delta$ loss gap:

$$l(i, k_2) \geq l(i, k_1) + \delta$$

and

$$l(j, k_2) \leq l(j, k_1) - \delta$$

Thus:

$$\|L_i - L_j\|^2 \geq (l(i, k_1) - l(j, k_1))^2 + (l(i, k_1) - l(j, k_1) + 2\delta)^2$$

The right side is minimized when $(l(i, k_1) - l(j, k_1))^2 = (l(i, k_1) - l(j, k_1) + 2\delta)^2$, or when $l(i, k_1) - l(j, k_1) = -\delta$. Thus:

$$\|L_i - L_j\|^2 \geq 2\delta^2$$

Hence, the result follows. □

Note that in *IFCA* the best model for each cluster is initialized to be close to its optimal counterpart. Consequently, $\delta$ can be much larger than $0$. Lemma A.17 therefore implies significant distance between the loss vectors of clients of different clusters.

# B   DISCUSSION

## B.1   COMMUNICATION AND COMPUTATION COSTS

Compared with other PFL methods like *IFCA* that exchange only model weights, *CLoVE* additionally sends the loss vectors. The size of the loss vectors is $\Theta(K)$ per client. This is small compared to the model weights themselves, which are $\Theta(K * R)$ per client, where $R$ is the number of model parameters (or gradients). Also, in practice, $K$ is usually small. So the overhead of communicating the loss vectors is not significant.

Furthermore, thanks to *CLoVE*'s early stopping feature, all models need only to be sent to all clients for the rounds until stability is reached. As shown in our experiments (see App. D.1.5), stability is achieved in a few rounds with high probability. After that, only a single model needs to be sent to each client. From then on, each cluster performs regular *FedAvg*, so the cost is similar to *FedAvg*, which is the best possible (other baselines also have a computation and communication cost at least as high as that of *FedAvg*).

Another technique for further reducing the size of the communicated models in the first few rounds is to leverage weight-sharing techniques from multi-task learning, as suggested also in *IFCA*: the $K$ models can share a few initial layers that are meant to learn the common properties of the data, and the rest of the layers are meant to learn the distinct features of each cluster. This way, when the server sends the models to each client, it needs to only send the common part once alongside the distinct parts, thus achieving savings in communication costs.

Finally, one might argue that *CLoVE* might result in high peak memory usage, due to clients evaluating all the models before cluster stability is reached. This can be alleviated as follows: the $K$ models do not need to be evaluated by the client all at the same time. Instead, if memory is a constraint, the client can be requesting and evaluating each of the $K$ models sequentially one at a time, storing in memory only the loss produced by each model. This way, it can form the loss vector while only holding one model in memory at a time.

## B.2  PRIVACY

Although privacy is not the focus of this paper, it is an important concern in FL and we now briefly discuss it. Compared to standard FL algorithms such as *FedAvg* that share model weights/gradients, clients in *CLoVE* additionally share the average model losses. Importantly, these loss values are not the raw model outputs; they are computed based on the output and the (unknown to the server) input or labels, and also averaged for all datapoints of the client before being sent to the server. While this aggregate measure is generally not considered privacy-sensitive, its privacy implications remain underexplored. Further research is needed to assess whether, and under what conditions, such aggregate statistics could lead to unintended information leakage, not with regard to *CLoVE* in particular, but more generally. Moreover, several baselines (e.g. Tan et al. (2022); Xu et al. (2023); Vahidian et al. (2023)) similarly share 1-D vectors or summaries of client data (we only send losses, not summaries of data). This process is deemed irreversible by those baselines, and they even mention that further privacy-preserving techniques can be employed.

## B.3  COMPARISON WITH FEDGWC PAPER

Licciardi et al. (2025) is a hierarchical approach like Sattler et al. (2021), and as such the number of rounds for clustering can be very large, as is also evident from the analytical guarantees that are only in the asymptotic regime as the number of rounds grows without bound; in contrast, *CLoVE* achieves correct clustering in 2-3 rounds, as shown both analytically and empirically. Moreover, in their approach, amount of state kept at server is quadratic in number of clients. With our approach, the amount of state at the server is not a function of the number of clients, but rather it is linear on the number of clusters, which is typically much smaller than number of clients. Additionally, FedGWC's clustering happens on the basis of losses computed on a single model. Specifically, the vectors used in Eq. 4 for computing entries of matrix $W$ are based on the losses (and rewards) of clients on the single model that is sent to the clients in that round. On the other hand, CLoVE performs clustering by comparing clients' losses across multiple models. This can be more effective, as losses of multiple models amplify the difference in data distributions among clients belonging to different clusters. This is why CLoVE is able to recover clusters so much faster.

## C  DETAILS ON DATASETS, MODELS AND COMPUTE USED

**Datasets**  We use the MNIST dataset (LeCun et al., 1998), containing handwritten digit images from 0 to 9 (10 classes; 60K points training, 10K testing), the CIFAR10 dataset (Krizhevsky, 2009), containing color images from 10 categories (airplane, bird, etc.) (50K points training, 10K testing), and the Fashion-MNIST (FMNIST) image dataset (Xiao et al., 2017), containing images of clothing items from 10 categories (T-shirt, dress, etc.) (60K points training, 10K testing). Beyond image datasets, we also use two text datasets: Amazon Review Data (Ni et al., 2019) (denoted by Amazon Review), containing Amazon reviews from 1996-2018 for products from 4 categories (2834 for books, 1199 for DVDs, 1883 for electronics, and 1755 for kitchen and houseware) and the consumer sentiment for each product based on the reviews (binary – positive/negative), and AG News (Gulli; Zhang et al., 2015), containing news articles (30000 training and 1900 testing) from 4 classes (world, sports, business, and science). Finally, we use the Federated EMNIST (FEMNIST) dataset (Caldas et al., 2019), an FL dataset with data points being handwritten letters or digits from a particular human writer. All these datasets are also used by the baselines.

**Models**  For MNIST and FMNIST unsupervised image reconstruction, we used a fully connected autoencoder with Sigmoid output activation. For CIFAR-10 unsupervised image reconstruction, we used a Convolutional Autoencoder with a fully connected latent bottleneck and a Tanh output activation. For supervised MNIST, FMNIST and FEMNIST image classification we used a simple convolutional neural network (CNN) classifier designed for grayscale images. Each convolutional layer uses: 5×5 kernels, Stride 1 (default) followed by ReLU and 2×2 max pooling. Its output is a logit vector. For CIFAR-10 image classification, we used a deep convolutional neural network classifier, designed for 32×32 RGB images. It uses batch normalization for stable training and dropout in fully connected layers to reduce overfitting. For AG News we used a TextCNN with an embedding layer, and for Amazon News we used a simple multi-layer perceptron (MLP) model.

Note that we are the first to propose an algorithm that can cluster correctly *using autoencoders in a fully unsupervised setting*. The datasets we use can all be clustered successfully using simple models, thanks to *CLoVE*'s superior ability to separate the clients, so there is no need to employ heavier models for these tasks. (If a task requires a certain type of model, *CLoVE* should again work well, as it is only based on the loss that the model outputs and the model itself is used as a black box).

**Experiment setup**   We provide the details on the number of clients and datapoints used for each of our experiments in Table 6.

Table 6: Details on number of clients and datapoints (S: Supervised, U: Unsupervised)

| Experiment | Number of clients | | | Number of datapoints per client | | |
|---|---|---|---|---|---|---|
| | MNIST | CIFAR-10 | FMNIST | MNIST | CIFAR-10 | FMNIST |
| S: Label skew 1 | 25 | 15 | 25 | 500 | 500 | 500 |
| S: Label skew 2 | 25 | 15 | 25 | 500 | 500 | 500 |
| S: Label skew 3 | 25 | 25 | 25 | 500 | 500 | 500 |
| S: Label skew 4 | 50 | 30 | 50 | 500 | 500 | 500 |
| S: Feature skew | 40 | 20 | 40 | 500 | 500 | 500 |
| S: Concept shift | 20 | 12 | 20 | 1000 | 1000 | 1000 |
| S: Amazon Review | | 20 | | | 1000 | |
| S: AG News | | 30 | | | 100 | |
| S: Ablation (CIFAR-10) | | 25 | | | 500 | |
| U: Initialization (MNIST) | | 50 | | | 500 | |
| U: Linear | | 25 | | | 1000 | |
| U: Everything else | | 50 | | | 1000 | |

**GitHub links**   The GitHub links for the code for each of the baseline algorithms used in the paper are shown in Table 7.

Table 7: GitHub links for comparison algorithms

| Algorithm | Source code |
|---|---|
| Per-FedAvg | `https://github.com/TsingZ0/PFLlib` |
| FedProto | `https://github.com/TsingZ0/PFLlib` |
| FedALA | `https://github.com/TsingZ0/PFLlib` |
| FedPAC | `https://github.com/TsingZ0/PFLlib` |
| CFL-S | `https://github.com/felisat/clustered-federated-learning` |
| FeSEM | `https://github.com/morningD/FlexCFL` |
| FlexCFL | `https://github.com/morningD/FlexCFL` |
| PACFL | `https://github.com/MMorafah/PACFL` |
| FedAvg | self-implemented |
| Local-only | self-implemented |
| IFCA | self-implemented |
| CLoVE | self-implemented |

**Compute used**   We train the neural networks and run all our evaluations on a server with 98 cores with Intel(R) Xeon(R) Gold 6248R CPU @ 3.00GHz, and 256GB RAM. For our experiments, we used Python 3.12 and PyTorch (Paszke et al., 2019), version 2.7.0.

# D ADDITIONAL EXPERIMENTAL RESULTS

## D.1 ADDITIONAL SUPERVISED RESULTS

### D.1.1 MIXED LINEAR REGRESSION EXPERIMENTS: COMPARISON WITH K-FED

We conduct a study to evaluate the impact of closeness of the different clusters optimal model parameters on *CLoVE*'s clustering performance. We select a supervised setting, where each cluster's data is generated by a linear model, as described in Sec. 4. We randomly select optimal model parameters $\theta_k^* \in \mathbb{R}^d$ from a unit $d$-dimensional sphere, ensuring that the pairwise distance between them falls within the range $[\Delta, 5\Delta]$. Our experimental setup consists of $K = 5$ clusters and $M = 25$ clients; each client has $n = 1000$ data points, and we use $\tau = 25$ local epochs. Figure 3(a) illustrates the clustering accuracy of *CLoVE* over multiple rounds, with $\Delta$ varying from 0.0003 to 1.0.

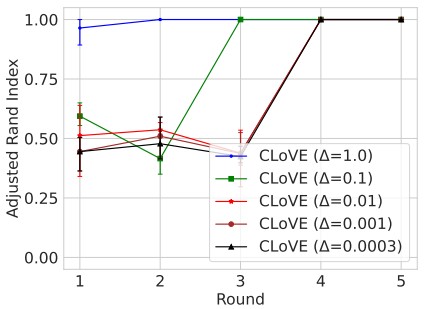

(a) Clustering accuracy of *CLoVE* over time (rounds) for different values of $\Delta$

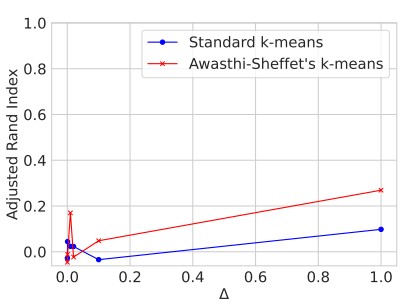

(b) Clustering accuracy of *k-FED* using standard k-means vs with Awasthi & Sheffet (2012)'s enhancements

Figure 3: Linear regression experiment results

As shown, *CLoVE* consistently recovers the correct cluster assignment. Moreover, as expected, the convergence rate of *CLoVE* slows down when the optimal model parameters are closer together (i.e. when $\Delta$ is smaller). In contrast, the performance of *k-FED* (Dennis et al., 2021) on the same data, even with Awasthi-Sheffet's k-means enhancements (Awasthi & Sheffet, 2012), is significantly worse (Fig. 3(b)). This discrepancy can be attributed to the fact that the cluster centers (i.e. the mean of their data) are very close to the origin, highlighting the limitations of *k-FED* in such scenarios. Our results demonstrate that *CLoVE* outperforms *k-FED*, especially when the cluster centers are not well-separated.

## D.1.2 ADDITIONAL LABEL SKEWS

Table 8 presents image classification test accuracy results for MNIST, FMNIST, CIFAR-10 datasets on additional types of label skews. The corresponding ARI accuracy results are presented in Table 9 and the clustering convergence results in Table 10. These results include two cases *Label skew 3* and *Label skew 4*. The case *Label skew 3* is a data distribution in which there is significant overlap between labels assigned to clients of different clusters. Specifically, there are 5 clusters with 5 clients each. Among the first 4 clusters, all clients of a cluster get samples from all 10 classes except one. The missing class is uniquely selected for each of the 4 clusters. The clients of the 5th cluster get samples from all 10 classes. The case *Label skew 4* is a data distribution in which $s\%$ of data (we use one third) for clients belonging to a cluster is uniformly sampled from all classes, and the remaining $(100 - s)\%$ comes from a dominant class that is unique for each cluster (Xu et al., 2023). Each cluster has 5 clients each of which is assigned 500 samples. For *Label skew 3*, the data of a class is distributed amongst the clients that are assigned that class by sampling from a Dirichlet distribution with parameter $\alpha = 0.5$, according to the procedure described in Section 5.

We observe that in some cases for *Label skew 3* and *Label skew 4*, *FedAvg* is achieving the highest test accuracy. This can be attributed to the fact that under these cases, each client holds data from almost every data class, which makes the client distributions very similar. Thus, naturally, the single

model that *FedAvg* trains is also a good model for the individual clients. Moreover, the model of *FedAvg* is trained on the data of all clients, while the other algorithms only use the data of a single cluster to train each of their models. This gives *FedAvg* an advantage, as it trained on much more data for the same number of rounds.

Table 8: Supervised test accuracy results for MNIST, CIFAR-10 and FMNIST

| Data mixing | Algorithm | MNIST | CIFAR-10 | FMNIST |
|---|---|---|---|---|
| | FedAvg | $\mathbf{99.0 \pm 0.0}$ | $\mathbf{84.1 \pm 0.5}$ | $\mathbf{88.8 \pm 0.3}$ |
| | Local-only | $85.9 \pm 0.6$ | $41.1 \pm 1.5$ | $71.1 \pm 0.4$ |
| Label skew 3 | Per-FedAvg | $95.3 \pm 0.3$ | $41.6 \pm 0.7$ | $74.9 \pm 0.7$ |
| | FedProto | $88.8 \pm 0.2$ | $39.9 \pm 0.4$ | $74.1 \pm 0.2$ |
| | FedALA | $94.7 \pm 0.3$ | $40.0 \pm 0.7$ | $75.1 \pm 0.5$ |
| | FedPAC | $95.6 \pm 0.2$ | $41.4 \pm 0.6$ | $75.5 \pm 0.8$ |
| | CFL-S | $97.9 \pm 0.1$ | $61.2 \pm 0.6$ | $84.3 \pm 1.2$ |
| | FeSEM | $94.0 \pm 0.2$ | $13.3 \pm 0.2$ | $72.2 \pm 1.9$ |
| | FlexCFL | $92.2 \pm 0.3$ | $20.8 \pm 0.5$ | $79.7 \pm 0.5$ |
| | PACFL | $82.8 \pm 0.9$ | $27.1 \pm 3.8$ | $68.1 \pm 2.1$ |
| | IFCA | $94.2 \pm 0.2$ | $57.3 \pm 1.9$ | $80.6 \pm 1.0$ |
| | CLoVE | $94.2 \pm 0.2$ | $56.9 \pm 0.5$ | $79.5 \pm 0.5$ |
| | FedAvg | $\mathbf{98.5 \pm 0.1}$ | $60.7 \pm 0.8$ | $85.9 \pm 0.2$ |
| | Local-only | $94.8 \pm 0.2$ | $67.1 \pm 0.5$ | $85.5 \pm 0.1$ |
| Label skew 4 | Per-FedAvg | $95.6 \pm 0.1$ | $63.7 \pm 0.8$ | $83.7 \pm 0.1$ |
| | FedProto | $93.3 \pm 0.2$ | $45.3 \pm 0.7$ | $76.5 \pm 0.3$ |
| | FedALA | $96.8 \pm 0.1$ | $69.8 \pm 0.3$ | $87.3 \pm 0.1$ |
| | FedPAC | $96.1 \pm 0.2$ | $66.3 \pm 0.5$ | $84.8 \pm 0.3$ |
| | CFL-S | $97.8 \pm 0.1$ | $71.3 \pm 1.1$ | $88.2 \pm 0.4$ |
| | FeSEM | $94.9 \pm 0.3$ | $11.9 \pm 1.5$ | $80.6 \pm 1.1$ |
| | FlexCFL | $97.2 \pm 0.1$ | $66.8 \pm 0.0$ | $89.5 \pm 0.1$ |
| | PACFL | $93.4 \pm 0.2$ | $67.4 \pm 0.3$ | $84.1 \pm 0.2$ |
| | IFCA | $98.0 \pm 0.1$ | $64.3 \pm 0.9$ | $89.4 \pm 0.2$ |
| | CLoVE | $97.8 \pm 0.2$ | $\mathbf{72.6 \pm 0.3}$ | $\mathbf{89.9 \pm 0.1}$ |

Table 9: Supervised ARI results for MNIST, CIFAR-10 and FMNIST

| Data mixing | Algorithm | MNIST | CIFAR-10 | FMNIST |
|---|---|---|---|---|
| Label skew 3 | CFL-S | $0.00 \pm 0.00$ | $0.77 \pm 0.18$ | $0.00 \pm 0.00$ |
| | FeSEM | $0.18 \pm 0.00$ | $0.00 \pm 0.00$ | $0.00 \pm 0.00$ |
| | FlexCFL | $\mathbf{1.00 \pm 0.00}$ | $\mathbf{1.00 \pm 0.00}$ | $\mathbf{1.00 \pm 0.00}$ |
| | PACFL | $0.20 \pm 0.08$ | $0.00 \pm 0.00$ | $0.15 \pm 0.13$ |
| | IFCA | $0.92 \pm 0.12$ | $0.69 \pm 0.08$ | $0.65 \pm 0.15$ |
| | CLoVE | $\mathbf{1.00 \pm 0.00}$ | $\mathbf{1.00 \pm 0.00}$ | $\mathbf{1.00 \pm 0.00}$ |
| Label skew 4 | CFL-S | $0.00 \pm 0.00$ | $0.61 \pm 0.03$ | $0.29 \pm 0.02$ |
| | FeSEM | $0.12 \pm 0.00$ | $0.00 \pm 0.00$ | $0.11 \pm 0.02$ |
| | FlexCFL | $\mathbf{1.00 \pm 0.00}$ | $\mathbf{1.00 \pm 0.00}$ | $\mathbf{1.00 \pm 0.00}$ |
| | PACFL | $0.53 \pm 0.06$ | $0.66 \pm 0.07$ | $0.92 \pm 0.02$ |
| | IFCA | $0.55 \pm 0.03$ | $0.36 \pm 0.22$ | $0.50 \pm 0.05$ |
| | CLoVE | $0.60 \pm 0.10$ | $\mathbf{1.00 \pm 0.00}$ | $0.95 \pm 0.07$ |

Table 10: Convergence behavior for MNIST, CIFAR-10 and FMNIST

| Data mixing | Algorithm | ARI reached in 10 rounds | | | First round when ARI $\geq 0.9$ | | |
|---|---|---|---|---|---|---|---|
| | | MNIST | CIFAR-10 | FMNIST | MNIST | CIFAR-10 | FMNIST |
| Label skew 2 | CFL-S | $0.00 \pm 0.00$ | $0.00 \pm 0.00$ | $0.00 \pm 0.00$ | — | — | — |
| | FeSEM | $0.18 \pm 0.00$ | $0.00 \pm 0.00$ | $0.00 \pm 0.00$ | — | — | — |
| | FlexCFL | $\mathbf{1.00 \pm 0.00}$ | $\mathbf{1.00 \pm 0.00}$ | $\mathbf{1.00 \pm 0.00}$ | $\mathbf{1.0 \pm 0.0}$ | $\mathbf{1.0 \pm 0.0}$ | $\mathbf{1.0 \pm 0.0}$ |
| | PACFL | $0.35 \pm 0.12$ | $0.32 \pm 0.11$ | $0.55 \pm 0.03$ | — | — | — |
| | IFCA | $0.83 \pm 0.12$ | $0.72 \pm 0.00$ | $0.92 \pm 0.12$ | — | — | — |
| | CLoVE | $\mathbf{1.00 \pm 0.00}$ | $\mathbf{1.00 \pm 0.00}$ | $\mathbf{1.00 \pm 0.00}$ | $2.0 \pm 0.0$ | $2.0 \pm 0.0$ | $2.0 \pm 0.0$ |
| Label skew 3 | CFL-S | $0.00 \pm 0.00$ | $0.00 \pm 0.00$ | $0.00 \pm 0.00$ | — | — | — |
| | FeSEM | $0.18 \pm 0.00$ | $0.00 \pm 0.00$ | $0.00 \pm 0.00$ | — | — | — |
| | FlexCFL | $\mathbf{1.00 \pm 0.00}$ | $\mathbf{1.00 \pm 0.00}$ | $\mathbf{1.00 \pm 0.00}$ | $\mathbf{1.0 \pm 0.0}$ | $\mathbf{1.0 \pm 0.0}$ | $\mathbf{1.0 \pm 0.0}$ |
| | PACFL | $0.20 \pm 0.08$ | $0.00 \pm 0.00$ | $0.15 \pm 0.13$ | — | — | — |
| | IFCA | $0.92 \pm 0.12$ | $0.69 \pm 0.08$ | $0.65 \pm 0.15$ | — | — | — |
| | CLoVE | $\mathbf{1.00 \pm 0.00}$ | $\mathbf{1.00 \pm 0.00}$ | $\mathbf{1.00 \pm 0.00}$ | $2.0 \pm 0.0$ | $2.0 \pm 0.0$ | $2.0 \pm 0.0$ |
| Label skew 4 | CFL-S | $0.00 \pm 0.00$ | $0.00 \pm 0.00$ | $0.00 \pm 0.00$ | — | — | — |
| | FeSEM | $0.12 \pm 0.00$ | $0.00 \pm 0.00$ | $0.11 \pm 0.02$ | — | — | — |
| | FlexCFL | $\mathbf{1.00 \pm 0.00}$ | $\mathbf{1.00 \pm 0.00}$ | $\mathbf{1.00 \pm 0.00}$ | $\mathbf{1.0 \pm 0.0}$ | $\mathbf{1.0 \pm 0.0}$ | $\mathbf{1.0 \pm 0.0}$ |
| | PACFL | $0.53 \pm 0.06$ | $0.66 \pm 0.07$ | $0.92 \pm 0.02$ | — | — | — |
| | IFCA | $0.54 \pm 0.03$ | $0.70 \pm 0.12$ | $0.50 \pm 0.05$ | — | — | — |
| | CLoVE | $0.90 \pm 0.07$ | $\mathbf{1.00 \pm 0.00}$ | $\mathbf{1.00 \pm 0.00}$ | $2.0 \pm 0.0$ | $2.0 \pm 0.0$ | $2.0 \pm 0.0$ |
| Feature skew | CFL-S | $0.00 \pm 0.00$ | $0.00 \pm 0.00$ | $0.00 \pm 0.00$ | — | — | — |
| | FeSEM | $0.00 \pm 0.00$ | $0.00 \pm 0.00$ | $0.00 \pm 0.00$ | — | — | — |
| | FlexCFL | $\mathbf{1.00 \pm 0.00}$ | $0.11 \pm 0.08$ | $\mathbf{1.00 \pm 0.00}$ | $\mathbf{1.0 \pm 0.0}$ | — | $\mathbf{1.0 \pm 0.0}$ |
| | PACFL | $0.00 \pm 0.00$ | $0.00 \pm 0.00$ | $0.68 \pm 0.11$ | — | — | — |
| | IFCA | $0.67 \pm 0.28$ | $0.71 \pm 0.22$ | $0.55 \pm 0.10$ | — | — | — |
| | CLoVE | $\mathbf{1.00 \pm 0.00}$ | $\mathbf{1.00 \pm 0.00}$ | $\mathbf{1.00 \pm 0.00}$ | $2.0 \pm 0.0$ | $4.3 \pm 0.9$ | $2.0 \pm 0.0$ |

Table 11: Supervised ARI results for Amazon Review and AG News

| Algorithm | Amazon Review | AG News |
|---|---|---|
| CFL-S | $0.07 \pm 0.08$ | $0.39 \pm 0.07$ |
| FeSEM | $0.00 \pm 0.01$ | $0.14 \pm 0.00$ |
| FlexCFL | $0.00 \pm 0.02$ | $0.02 \pm 0.00$ |
| PACFL | $0.00 \pm 0.00$ | $0.00 \pm 0.00$ |
| IFCA | $0.29 \pm 0.20$ | $0.64 \pm 0.12$ |
| CLoVE | $\mathbf{0.73 \pm 0.24}$ | $\mathbf{0.94 \pm 0.09}$ |

### D.1.3 ADDITIONAL DETAILS FOR TEXT CLASSIFICATION

The ARI accuracy and speed of convergence results for the Amazon Review and AG News datasets are presented in Table 11 and Table 12, respectively.

### D.1.4 THE CASE OF UNKNOWN NUMBER OF CLUSTERS $K$

As mentioned in Section 3, *CLoVE* does not necessarily require the true number of clusters to be known in advance and given as an input, but it can converge to an appropriate number of clusters during its execution. To demonstrate this feature of *CLoVE*, we perform the following experiment on the FEMNIST dataset. We use $M = 100$ clients and assign to each client all of the data from one of the human writers in FEMNIST. We use a CNN and random first-round weight initialization. In this case, the number of true clusters (which human writers have a similar writing style) is unknown. *CLoVE* searches over different possibilities for the number of clusters/models, achieving an accuracy of $68.2 \pm 1.3$. This demonstrates the versatility of our algorithm in general settings where the number of true clusters is not given as input.

Table 12: Convergence behavior for the Amazon Review and AG News datasets

| Algorithm | ARI reached in 10 rounds | | First round when ARI $\geq 0.9$ | |
|---|---|---|---|---|
| | Amazon | AG News | Amazon | AG News |
| CFL-S | $0.00 \pm 0.00$ | $0.00 \pm 0.00$ | — | — |
| FeSEM | $0.00 \pm 0.01$ | $0.14 \pm 0.00$ | — | — |
| FlexCFL | $0.00 \pm 0.02$ | $0.02 \pm 0.00$ | — | — |
| PACFL | $0.00 \pm 0.00$ | $0.00 \pm 0.00$ | — | — |
| IFCA | $0.29 \pm 0.20$ | $0.64 \pm 0.12$ | — | — |
| CLoVE | $\mathbf{0.72 \pm 0.04}$ | $\mathbf{0.94 \pm 0.09}$ | — | $\mathbf{2.0 \pm 0.0}$ |

### D.1.5 EARLY STOPPING OF CLUSTERING

As mentioned in Section 3, *CLoVE* has the ability to stop clustering (and the associated sending of all models by the server to each client and the inference by each client on all the models to generate the loss vectors) early once it stabilizes, so as to save in communication and computation costs. In this section, we demonstrate the fact that *CLoVE* indeed can reach a stable clustering within a few rounds. We run the same supervised classification experiments on CIFAR-10 for different modes of data mixing as before, but now allowing early stopping when the clustering stabilizes (we define stability as 3 rounds of no change). In all experiments, *CLoVE* achieves an ARI of 1.0. The results for the test accuracy and the round at which cluster stability is reached are shown in Table 13. We observe that *CLoVE* is able to reach a stable client-to-cluster assignment within 4-7 rounds in all scenarios.

Table 13: Early stopping experiments (CIFAR-10)

| Data mixing | Test accuracy | Round of stability |
|---|---|---|
| Label skew 1 | $90.7 \pm 0.2$ | $4.0 \pm 0.0$ |
| Label skew 2 | $67.2 \pm 1.5$ | $4.0 \pm 0.0$ |
| Feature skew | $58.2 \pm 1.2$ | $6.3 \pm 1.2$ |
| Concept shift | $59.4 \pm 0.4$ | $4.0 \pm 0.0$ |

### D.1.6 PARTIAL PARTICIPATION

We now experiment with different levels of client participation (parameter $\rho$). We use MNIST in a supervised setting with a CNN, $K = 5$ clusters, $M = 125$ clients (25 per cluster), $n = 100$ datapoints/client, and random initialization of the model weights. The results are shown in Table 14. Note that for partial participation the ARI at each round is measured using only the clients that participated in that round and comparing their clustering with the ground truth clustering projected on those clients. We see that *CLoVE* very quickly and consistently achieves perfect ARI and almost perfect accuracy even for very small (e.g. 10%) client participation rates.

Table 14: *CLoVE*'s performance under partial participation

| Participation rate $\rho$ | Test accuracy | Final ARI | ARI reached in 10 rounds | First round when ARI $\geq 0.9$ |
|---|---|---|---|---|
| 1.0 | $99.5 \pm 0.1$ | $1.00 \pm 0.00$ | $1.00 \pm 0.00$ | $2.0 \pm 0.0$ |
| 0.9 | $99.6 \pm 0.1$ | $1.00 \pm 0.00$ | $1.00 \pm 0.00$ | $2.0 \pm 0.0$ |
| 0.75 | $99.4 \pm 0.2$ | $1.00 \pm 0.00$ | $1.00 \pm 0.00$ | $2.0 \pm 0.0$ |
| 0.5 | $99.6 \pm 0.1$ | $1.00 \pm 0.00$ | $1.00 \pm 0.00$ | $2.0 \pm 0.0$ |
| 0.25 | $99.4 \pm 0.1$ | $1.00 \pm 0.00$ | $1.00 \pm 0.00$ | $2.0 \pm 0.0$ |
| 0.1 | $99.6 \pm 0.2$ | $1.00 \pm 0.00$ | $1.00 \pm 0.00$ | $3.0 \pm 0.8$ |

We also perform an experiment for the extreme case where participation is so low that the number of participating clients is lower than the true number of clusters. We examine *CLoVE*'s behavior with early stopping after cluster stability (Fig. 4(a)) and without early stopping (Fig. 4(b)). We use MNIST with $K = 10$ clusters, $m = 2$ clients per cluster and $\rho = 40\%$ participation rate (so only 8 clients

participate at each round, which is lower than the true number of clusters, 10). The ARI is evaluated only for clients that have participated at least once in the training process until that round. In the first few rounds, the ARI is higher, then it drops, and then again rises to the optimal. This is because in the beginning very few clients have participated until that point, so there are not many wrong cluster assignments. As more clients join, initially the number of incorrect assignments increases, but then, as *CLoVE* enhances the assignments over time, the ARI increases once again. As can be seen, after a few tens of rounds, *CLoVE* manages to almost reach optimal clustering, even though this setting is very challenging.

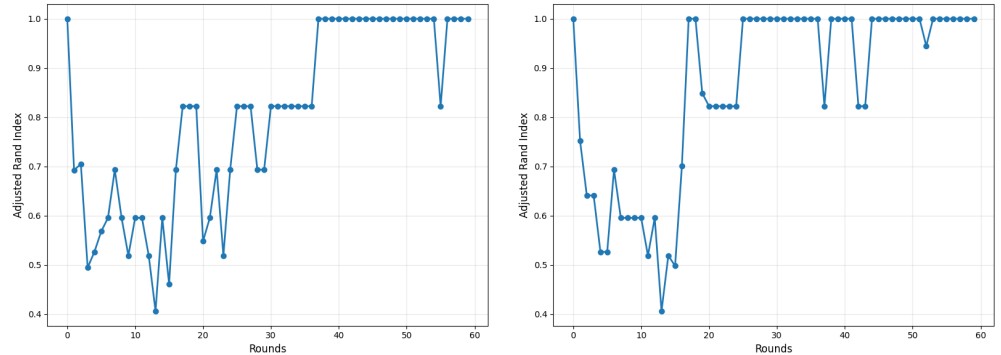

Figure 4: ARI for very few participants – fewer than the number of clusters – with early stopping (left) and without early stopping (right).

### D.2 ADDITIONAL UNSUPERVISED RESULTS

For this section's experiments, we have used a batch size of 64.

#### D.2.1 BENEFIT OF PERSONALIZATION

We now examine the reconstruction loss performance of the trained models and show the benefit of our personalized models over traditional FL models. For MNIST and $K = 10$, $M = 50$ and $n = 250$, we run 4 different algorithms: *CLoVE*, *IFCA*, *FedAvg*, *Local-only*. In *FedAvg*, i.e. the original FL algorithm, a single model is trained and aggregated by all clients, regardless of their true cluster (equivalently, *FedAvg* is *CLoVE* with $K = 1$). *Local-only* means federated models but no aggregation: each client trains its own model only on its local data, and does not share it with the rest of the system.

For this experiment, we consider that each of the 50 clients has a testset of the same data class of the client's trainset, and we feed its testset through all models (10 models for *CLoVE* and *IFCA*, 1 for *FedAvg* and 50 for *Local-only*), creating the loss matrices shown as heatmaps in Fig. 5 (run for a single seed value). The vertical axis represents clients, one selected from each true cluster, and the horizontal represents the different models. Clients and models are labeled so that the diagonal represents the true client to model/cluster assignment. Numbers on the heatmap express the reconstruction loss (multiplied by a constant for the sake of presentation) for each client testset and model combination. Very high values are hidden and the corresponding cells are colored red. For the local case, we only show 10 of the 50 models, the ones that correspond to the selected clients.

We observe that in all cases, the reconstruction loss of each client's assigned model by *CLoVE* (the diagonal) is the smallest in the client's row (i.e. the smallest among all models), which confirms that the models *CLoVE* assigned to each client are actually trained to reconstruct the client's data (MNIST digit) well, and not reconstruct well the other digits. On the contrary, *IFCA* gets stuck and assigns all clients to 3 clusters (models) – the first 3 columns. The rest of the models remain practically untouched (untrained), hence their high (red) loss values on all clients' data. The 3 chosen models are not good for the reconstruction of MNIST digits, because they don't achieve the lowest loss on their respective digits.

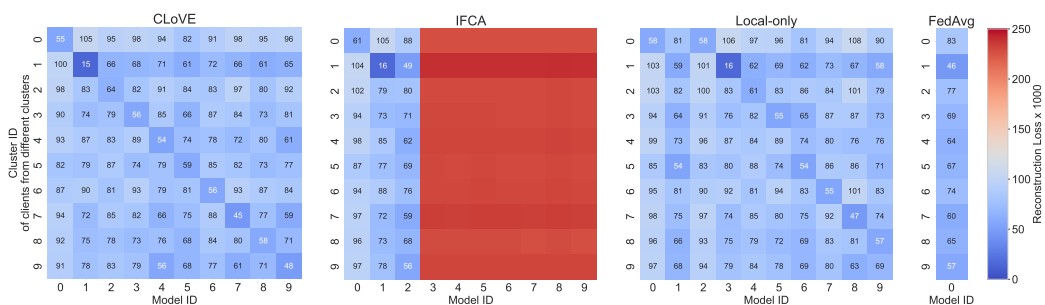

Figure 5: Heatmaps of reconstruction loss of MNIST digits by models clustered and trained by each algorithm

Comparing *CLoVE* and *FedAvg*, we see that for each row (client from each cluster), the loss achieved by *CLoVE* is lower than that achieved by *FedAvg*. This demonstrates the **benefit of our personalized FL approach over traditional FL**: *FedAvg* uses a single model to train on local data of each client, and aggregates updated parameters only for this single model. *CLoVE*, on the other hand, clusters the similar clients together and trains a separate model on the data of clients belonging to each cluster, thus effectively learning – and tailoring the model to – the cluster's particular data distribution. Finally, comparing *CLoVE* with *Local-only*, we see that the diagonal losses achieved by *CLoVE* on the selected models for each client are lower than the corresponding diagonal losses by *Local-only*. This shows the benefit of federated learning, as *CLoVE* models are trained on data from multiple clients from the same cluster, unlike *Local-only* where clients only train models based on their limited amount of private data.

### D.2.2 MODEL AVERAGING VERSUS GRADIENT AVERAGING

In this experiment, we compare the two variants of our *CLoVE* algorithm, i.e. *model averaging* and *gradient averaging*. We run an MNIST experiment with $K = 10$, $M = 50$ and $n = 100$, using both averaging methods. The results are shown in Fig. 6. We observe that *CLoVE* is able to arrive at the correct clustering with both methods, with gradient averaging needing slightly more rounds to do so than model averaging.

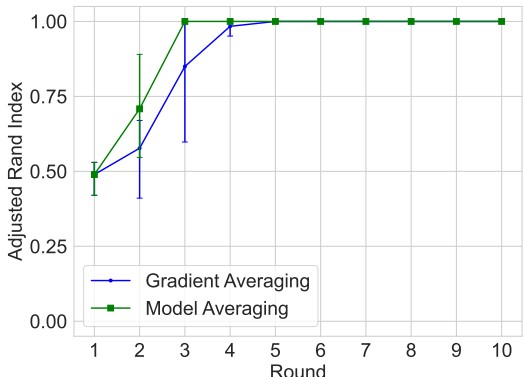

Figure 6: Clustering accuracy (ARI) of *CLoVE* over time (rounds) for MNIST for model averaging vs gradient averaging.

## D.3 SCALING WITH THE NUMBER OF CLIENTS

In this experiment, we examine the effect of increasing the number of clients. We use 10 clusters, and 100-200 datapoints per client. The results for MNIST, CIFAR-10 and FMNIST in the supervised setting (using CNNs) are shown in Table 15, and for MNIST in the unsupervised setting (using autoencoders) are shown in Table 16. In the supervised setting, we see that *CLoVE*'s final ARI is always perfect, and both the ARI and the test accuracy of *CLoVE* consistently exceed those of *IFCA*, no matter how big the number of clients is. Similarly, in the unsupervised setting we observe that *IFCA* consistently achieves a bad ARI, while *CLoVE* always manages to reach perfect clustering accuracy and scales successfully when the number of clients is large.

Table 15: Scaling with the number of clients – Supervised setting

| Algorithm | # of clients $M$ | Final ARI | | | Test accuracy | | |
|---|---|---|---|---|---|---|---|
| | | MNIST | CIFAR-10 | FMNIST | MNIST | CIFAR-10 | FMNIST |
| IFCA | 50 | $0.92 \pm 0.11$ | $0.85 \pm 0.11$ | $0.74 \pm 0.22$ | $99.2 \pm 0.8$ | $83.7 \pm 5.3$ | $96.8 \pm 2.3$ |
| | 100 | $0.85 \pm 0.11$ | $0.65 \pm 0.25$ | $0.93 \pm 0.11$ | $99.1 \pm 0.5$ | $72.7 \pm 12.7$ | $97.4 \pm 2.6$ |
| | 250 | $0.87 \pm 0.18$ | $0.85 \pm 0.10$ | $0.85 \pm 0.10$ | $99.4 \pm 0.7$ | $80.9 \pm 7.7$ | $98.6 \pm 0.5$ |
| | 500 | $0.93 \pm 0.10$ | $0.85 \pm 0.10$ | $0.85 \pm 0.10$ | $97.1 \pm 2.9$ | $77.3 \pm 5.8$ | $89.2 \pm 7.7$ |
| | 1000 | $0.85 \pm 0.10$ | $0.93 \pm 0.10$ | $0.75 \pm 0.21$ | $98.3 \pm 1.0$ | $78.0 \pm 7.2$ | $92.0 \pm 3.6$ |
| CLoVE | 50 | $1.00 \pm 0.00$ | $1.00 \pm 0.00$ | $1.00 \pm 0.00$ | $99.8 \pm 0.0$ | $90.8 \pm 0.2$ | $99.1 \pm 0.0$ |
| | 100 | $1.00 \pm 0.00$ | $1.00 \pm 0.00$ | $1.00 \pm 0.00$ | $99.8 \pm 0.0$ | $90.3 \pm 0.6$ | $99.3 \pm 0.0$ |
| | 250 | $1.00 \pm 0.00$ | $1.00 \pm 0.00$ | $1.00 \pm 0.00$ | $99.9 \pm 0.0$ | $91.2 \pm 1.5$ | $99.3 \pm 0.0$ |
| | 500 | $1.00 \pm 0.00$ | $1.00 \pm 0.00$ | $1.00 \pm 0.00$ | $99.6 \pm 0.1$ | $82.3 \pm 1.5$ | $95.3 \pm 4.4$ |
| | 1000 | $1.00 \pm 0.00$ | $1.00 \pm 0.00$ | $1.00 \pm 0.00$ | $99.6 \pm 0.1$ | $84.6 \pm 1.2$ | $96.5 \pm 3.0$ |

Table 16: Scaling with the number of clients – Unsupervised setting

| Algorithm | # of clients $M$ | Test loss | Final ARI |
|---|---|---|---|
| IFCA | 100 | $0.055 \pm 0.001$ | $0.091 \pm 0.064$ |
| | 200 | $0.055 \pm 0.001$ | $0.079 \pm 0.060$ |
| | 500 | $0.059 \pm 0.002$ | $0.210 \pm 0.149$ |
| | 1000 | $0.067 \pm 0.002$ | $0.186 \pm 0.137$ |
| CLoVE | 100 | $0.065 \pm 0.001$ | $1.000 \pm 0.000$ |
| | 200 | $0.063 \pm 0.000$ | $1.000 \pm 0.000$ |
| | 500 | $0.065 \pm 0.000$ | $1.000 \pm 0.000$ |
| | 1000 | $0.073 \pm 0.000$ | $1.000 \pm 0.000$ |

## D.4 CONVERGENCE DYNAMICS

We now show the convergence dynamics of *CLoVE* for MNIST in the unsupervised setting. We use $K = 10$ clusters, and $n = 100$ datapoints per client. We vary the number of clients from 50 to 200. The behavior of the ARI can be seen in Figure 7. We observe that *CLoVE* reaches a perfect ARI in just a few rounds, for all values of the number of clients $M$.

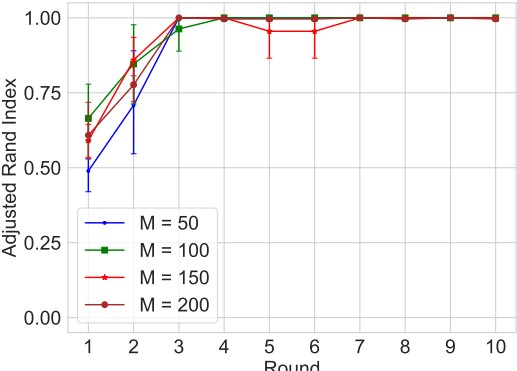

Figure 7: Clustering accuracy (ARI) of *CLoVE* over time (rounds) for MNIST for different numbers of clients

## D.5 ABLATION STUDIES

We analyze the impact of changing some of key components of *CLoVE* one at a time. This includes:

- **No Matching**: Order the clusters by their smallest client ID and assign models to them in that order (i.e. cluster $k$ gets assigned model $k$). This is in lieu of using the bi-partite matching approach of Alg. 1.
- **Agglomerative clustering**: Use a different method than k-means to cluster the loss vectors.
- **Square root loss**: Cluster vectors of square roots of model losses instead of the losses themselves.

Our experiments are for image classification for the CIFAR-10 dataset using 25 clients that are partitioned uniformly among 5 clusters using a label skew with very large overlap. The clients of the first four cluster are assigned samples from every class except one class which is chosen to be different for each cluster. The clients of the fifth cluster are assigned samples from all classes. The partitioning of the data of each class among the clients that are assigned to that class is by sampling from a Dirichlet($\alpha$) distribution with $\alpha = 0.5$. The test accuracy achieved is shown in Table 17. We find that "No Matching" has the most impact, as it reduces the test accuracy by almost $4.3\% \pm 1\%$. Switching to Agglomerative clustering results in a slight improvement in the mean test accuracy, but the result is not statistically significant. Likewise, square root loss results in no significant change in performance. This shows that *CLoVE* is robust in its design.

Table 17: Ablation studies

|  | Test accuracy |
| --- | --- |
| Original CLoVE | $58.3 \pm 1.4$ |
| No matching | $55.8 \pm 0.6$ |
| Agglomerative clustering | $59.3 \pm 1.5$ |
| Square root loss | $58.1 \pm 3.3$ |

