# OpenReview forum: "CLoVE: Personalized Federated Learning through Clustering of Loss Vector Embeddings"
_ICLR.cc/2026/Conference — Submitted to ICLR 2026_

### Official Review · Reviewer_3G9C · 2025-10-19

**Soundness:** 2
**Presentation:** 2
**Contribution:** 1
**Rating:** 2
**Confidence:** 4

**Summary:**

The paper proposes a clustered FL algorithm that groups clients in clusters based on the similarities of the local loss functions, instead of working directly on clients' models, using k-means on the loss embeddings and then assigning the models with The authors motivate their result by providing a theoretical analysis conducted on a simple linear regression scenario. The algorithm performance is assessed on classical datasets (mnist variants, cifar10) and extend to text datasets and compared with relevant clustered and personalized FL baselines.

**Strengths:**

The experimental evaluation compares their method with relevant CFL baselines, as well as with PFL algorithms. It is appreciable the theoretical comparison with IFCA in Appendix A3. Achieving early cluster stability is also an interesting property that permits to efficiently control the

**Weaknesses:**

1) Most of the theoretical claims, while complete, are limited to the linear regression case. This works well as a proof of concept, but further effort should be directed toward more complex and general cases. Providing a more general framework would definitely broaden the impact of the analysis. Furthermore, the mathematical soundness, clarity, and exposition could be improved.

2) The authors build upon an interesting line of work paved by Cho et al. (2022), focusing on the loss function to evaluate clusters. The premises and methodology are reminiscent of FedGWC (Licciardi et al., 2025). I suggest that the authors compare their method with this CFL approach, which is based on the transformation of loss vector embeddings and claims that clients can be grouped according to similarities in their learning patterns.

3) In FL protocols, it is usually preferable not to share private information with other clients (though some works are less stringent about this). My concern is that the algorithm lacks mechanisms to avoid degenerate clusters (i.e., a cluster with a single client). If this occurs, then during the broadcasting phase, every other client would gain access to the single client's model—and consequently, its data distribution—when cross-evaluating the loss functions. Furthermore, cross-evaluating models could be intractable in large-scale scenarios or in settings with full client participation.

4) The description of how the number of clusters, K(t+1), is determined is somewhat vague in both the pseudocode and the main text. Additionally, how does the min-cost bipartite matching step impact the efficiency of each communication round during the training phase?

5) To make the experimental framework more comparable with the standard FL literature, data heterogeneity should be controlled using a Dirichlet distribution. While the authors mention this in the appendix, clarifying this detail in the main experimental section would help the FL community better interpret the results.

6) The method achieves strong results in clustering quality, as measured by the ARI. However, its test accuracy generally underperforms compared to other baselines. Do the authors have any insights into this discrepancy and its potential causes?

7) I am a little doubtful on the algorithm's behaviour when the number of clusters is larger than the number of training clients per round -- something that could easily occur in a realistic framework. Does cluster optimality yield also in that scenario?

-------
- Cho, Yae Jee, Jianyu Wang, and Gauri Joshi. "Towards understanding biased client selection in federated learning." International Conference on Artificial Intelligence and Statistics. PMLR, 2022.
- Licciardi, A., Leo, D., Fanì, E., Caputo, B., & Ciccone, M. Interaction-Aware Gaussian Weighting for Clustered Federated Learning. In Forty-second International Conference on Machine Learning.

**Questions:**

Please, see weaknesses

---

> ### Author Response · Authors · 2025-11-27
>
> **W1: Most of the theoretical claims, while complete, are limited to the linear regression case. This works well as a proof of concept, but further effort should be directed toward more complex and general cases. Providing a more general framework would definitely broaden the impact of the analysis. Furthermore, the mathematical soundness, clarity, and exposition could be improved.**
>
> Thank you for the time and effort you put into reviewing our paper and for your insightful comments and suggestions.
>
> Our primary objective is to provide a rigorous analytical proof that CLoVE rapidly recovers the true client clusters. In particular, Theorem 4.1 shows that, with high probability, the correct clustering is achieved after a single iteration of CLoVE. However, as you have observed, we currently lack an analogous guarantee for the non convex setting despite our efforts in this direction. This gap reflects a fundamental difficulty: rapid convergence results in non convex settings are scarce in the literature. Existing baselines such as IFCA achieve fast convergence only when the underlying optimization problem is convex, and recent work (e.g., the ICML 2025 “FedGWC” paper that you point to, a discussion of which we added in a new Appendix B.3) provides recovery guarantees only in the asymptotic regime as the number of rounds grows without bound. Note that all of our empirical results (except for the ones in Appendix D.1.1 which is by design dedicated to the linear setting) are for the non-convex setting, which clearly shows that CLoVE is able to handle a variety of non-convex settings very successfully in practice. Providing formal guarantees in non-convex settings is a central focus of our planned future research.
>
> **W2: The authors build upon an interesting line of work paved by Cho et al. (2022), focusing on the loss function to evaluate clusters. The premises and methodology are reminiscent of FedGWC (Licciardi et al., 2025). I suggest that the authors compare their method with this CFL approach, which is based on the transformation of loss vector embeddings and claims that clients can be grouped according to similarities in their learning patterns.**
>
> Thank you for bringing the recent FedGWC paper to our attention. It is a hierarchical approach like CFL (Sattler et al.), and as such the number of rounds for clustering can be very large, as is also evident from the analytical guarantees that are only in the asymptotic regime as the number of rounds grows without bound; in contrast, our algorithm achieves correct clustering in 2-3 rounds, as shown both analytically and empirically. Moreover, in their approach, amount of state kept at server is quadratic in number of clients. With our approach, the amount of state at the server is not a function of the number of clients, but rather it is linear on the number of clusters, which is typically much smaller than number of clients. Additionally, FedGWC’s clustering happens on the basis of losses computed on a single model. Specifically, the vectors used in Eq. 4 for computing entries of matrix $W$ are based on the losses (and rewards) of clients on the single model that is sent to the clients in that round. On the other hand, CLoVE performs clustering by comparing clients’ losses across multiple models. This can be more effective, as losses of multiple models amplify the difference in data distributions among clients belonging to different clusters. This is why CLoVE is able to recover clusters so much faster.
>
> In the revised paper, we have added a citation to FedGWC, a mention in the Related Work section, as well as the above in-depth comparison in a newly added Appendix B.3.

---

> ### Author Response · Authors · 2025-11-27
>
> **W3: In FL protocols, it is usually preferable not to share private information with other clients (though some works are less stringent about this). My concern is that the algorithm lacks mechanisms to avoid degenerate clusters (i.e., a cluster with a single client). If this occurs, then during the broadcasting phase, every other client would gain access to the single client's model—and consequently, its data distribution—when cross-evaluating the loss functions. Furthermore, cross-evaluating models could be intractable in large-scale scenarios or in settings with full client participation.**
>
> Thank you for these comments. During the normal operation of CLoVE, there is no sharing of private information between clients. If though the degenerate cluster scenario happens as you describe, then firstly, what could be visible to the other clients would be the model of the single-client cluster, and not the client’s data itself. Secondly, clients do not know which other clients are in the same cluster as them or what the clustering is, and thus do not know if a certain cluster consists of only one client. Finally, note that a client being clustered on its own might even be desirable, as CLoVE would do that because this would be the best possible clustering to achieve the highest possible accuracy, which means that imposing a mechanism that would force at least two clients per cluster might hurt accuracy. If though this is important for some particular application, methods from the differential privacy literature could be applied (but despite this being a valid point, the baselines do not deal with it and it is beyond the scope of this paper).
>
> Regarding the intractability of cross-evaluation in large-scale scenarios: note that the amount of cross-evaluation that each client has to do is a function of the number of clusters, not the number of clients. In practical applications, the number of clusters is usually small. Additionally, this cross-evaluation only needs to happen during the first few rounds until cluster stability is achieved, as detailed in Appendix D.1.5 (“Early Stopping of Clustering”). From then on, each cluster essentially runs FedAvg. Also, if the memory of clients is constrained, our comment to Reviewer 8Ay6 can be applied: models can be sent from the server to the client and evaluated one at a time (we have added this remark in Appendix B.1 (“Communication and Computation Costs”) in the revised version). Please note that as far as full client participation is concerned, from each client’s point of view, the cost of model cross-evaluation does not change with the participation rate, as it is only a function of the number of clusters, and not the number of participating clients.

---

> ### Author Response · Authors · 2025-11-27
>
> **W4: The description of how the number of clusters, K(t+1), is determined is somewhat vague in both the pseudocode and the main text. Additionally, how does the min-cost bipartite matching step impact the efficiency of each communication round during the training phase?**
>
> Thank you for pointing out the need for this clarification. In our pseudocode we did not include the full details of how we pick $K^{(t+1)}$ due to space constraints and in order to improve the algorithm’s readability. In the description of the pseudocode, we explain that the choice of $K^{(t+1)}$ happens “via searching over a range of values and choosing the one that yields the highest silhouette score, or using the elbow method”. However, we realize that this is an important aspect and we modified the main text (Section 3) to include more details.
>
> In particular: If the number of clusters is unknown a priori, at each round, after the generation of the loss vectors, the server tries to find the best possible number of clusters $K$. The definition of the “best possible” happens by using e.g. the silhouette score: A range of candidate $K$ values is examined, up to a predetermined upper bound $\hat{K}$. For each candidate $K$, the loss vectors are clustered using Agglomerative Clustering (which does not require knowledge of the number of clusters). Then, the silhouette score of the resulting clustering is computed. Eventually, the value of $K$ with the highest silhouette score is returned. Note that this procedure is not necessary when the number of clusters $K$ is known a priori. We have added the pseudocode of this procedure as the new Algorithm 2, and its explanation in Section 3 of the revised paper.
>
> The role of the bipartite matching is to assign the existing models to the produced clusters, and to do so in a way that clusters generally get models for which they have low total loss. This is a problem that can be solved in polynomial time using known algorithms. E.g. for an upper bound $\hat{K}$ on the number of clusters/models, Jonker and Volgenant (1987) does it in in time $O(\hat{K}^3)$ (we added a citation of this paper in the main section). Given that this process only takes place at the server, which is usually not that constrained computation-wise compared to the clients, and also since the number of clusters is usually small in practical applications, the impact of bipartite matching on the overall efficiency of CLoVE is not significant.
>
> Jonker, R., Volgenant, A. A shortest augmenting path algorithm for dense and sparse linear assignment problems. Computing 38, 325–340 (1987). https://doi.org/10.1007/BF02278710
>
> **W5: To make the experimental framework more comparable with the standard FL literature, data heterogeneity should be controlled using a Dirichlet distribution. While the authors mention this in the appendix, clarifying this detail in the main experimental section would help the FL community better interpret the results.**
>
> Thank you for pointing out the need for this clarification. As you are saying, using a Dirichlet distribution is standard in the FL literature, and this is why we also adopted it for some of our data mixing experiments, and in particular for Label skews 1-3. This procedure partitions the data points of each class (or label) across the participating clients. For instance, suppose the clients are divided into three clusters and a particular class contains $L$ data points. First, we draw three proportions $p_1$, $p_2$, $p_3$ from a Dirichlet distribution with concentration parameter $\alpha = 0.5$ (by construction, $p_1 + p_2 + p_3 = 1$). Then we randomly select $Lp_1$ points of the class for the clients in cluster 1, $Lp_2$ points for the clients in cluster 2, and the remaining $Lp_3$ points for the clients in cluster 3. The selected points are assigned randomly among the clients within each respective cluster. We have added this clarification to the “Data Partitioning” paragraph of the main experimental section (Section 5) to enhance the readability of the paper.

---

> ### Author Response · Authors · 2025-11-27
>
> **W6: The method achieves strong results in clustering quality, as measured by the ARI. However, its test accuracy generally underperforms compared to other baselines. Do the authors have any insights into this discrepancy and its potential causes?**
>
> Thank you for the question. CLoVE is designed to excel on both the Adjusted Rand Index (ARI) and model accuracy, yet these objectives can sometimes conflict. In certain scenarios, clusters that yield a higher ARI correspond to a lower accuracy, and conversely, clusters with a modest ARI may produce superior accuracy. A concrete illustration of this tradeoff occurs under Label skew 3 (Tables 8 and 9), where FedAvg consistently outperforms every other CFL method on accuracy across all datasets, despite its poor clustering performance. In contrast, CLoVE typically achieves the highest ARI among the CFL algorithms. After recovering the cluster structure, CLoVE, like the other CFL approaches, applies standard FedAvg aggregation within each cluster. Although this procedure often identifies clusters with strong ARI, it does not automatically translate into higher accuracy. Nevertheless, a high ARI generally creates favorable conditions for accuracy, which explains why CLoVE’s accuracy remains competitive with, and frequently surpasses, that of the competing methods.
>
> Note that an advantage of CLoVE, beyond fast convergence, is that it is the only algorithm that **consistently** ranks among the top performers in almost all scenarios, even in challenging ones (like certain skews for CIFAR-10) where other algorithms fail.
>
> **W7: I am a little doubtful on the algorithm's behaviour when the number of clusters is larger than the number of training clients per round -- something that could easily occur in a realistic framework. Does cluster optimality yield also in that scenario?**
>
> Thank you for this observation. Although we believe this is a rare case in practice, we performed experiments for the extreme case where participation is so low that the number of participating clients is lower than the true number of clusters. Our experiments demonstrate that CLoVE achieves cluster optimality even in that scenario. Specifically, we examine CLoVE behavior with early stopping after cluster stability (Fig. 4, left) and without early stopping (Fig. 4, right). We use MNIST with $K = 10$ clusters, $m = 2$ clients per cluster and $\rho = 40\%$ participation rate (so only 8 clients participate at each round, which is lower than 10 - the true number of clusters). As can be seen from Figure 4, after a few tens of rounds, CLoVE still manages to almost reach optimal clustering, despite how challenging this setting is. We have added these additional experiments to Appendix D.1.6 (“Partial Participation”).

---

### Official Review · Reviewer_8Ay6 · 2025-10-29

**Soundness:** 3
**Presentation:** 3
**Contribution:** 3
**Rating:** 6
**Confidence:** 3

**Summary:**

This paper introduces CLoVE (Clustering of Loss Vector Embeddings), an algorithm for Clustered Federated Learning (CFL). The authors propose using loss vector embeddings as a basis for clustering clients, thereby addressing limitations in prior CFL approaches.

**Strengths:**

* Although similar to the prior method Iterative Federated Clustering Algorithm (IFCA), which uses the loss to build cluster identities, CLoVE simultaneously estimate the underlying clusters and constructs models per cluster. It does not require prescribe number of clusters and not sensitive to the model initialization, which consequently decides the cluster initialization.

* The manuscript is clearly written, methodologically solid, and well positioned within the current literature.

**Weaknesses:**

* The experiments choose some relative simple datasets to validate its effectiveness. For example, for minist, cifar-10, and FMNIST, several baseline models already achieve test 100\% accuracy. It's unclear how the methods perform on relative challenging datasets like Tiny-ImageNet.

* My concerns are on the communication costs and storage costs. Before the clustering stabilized, in each communication round, the server needs to broadcast $K^{(t)}$ models and the clients need to store them. Compared to the FedAvg, the communication costs may be averaged out if the stabilization happens during the early stages and the algorithm convergence faster than FedAvg. Theorem 4.1 may justify for the simple scenario, yet CLoVE's actually performance in recovering the clusters is unclear. Maybe showing the training dynamic can help to justify. Nonetheless, the peak memory usage can still be large. Even though IFCA method, also suffers from this. I think this should be clearly discussed.

**Questions:**

1. In Algorithm 1, what are the conditions to determine if the clustering is stabilized?

2. Is CLoVE scalable to large number of clients? The main paper mentioned that "We vary the number of clients between 20 and 1000." However, in appendix C, the number of clients ranging from 12 to 50. In the appendix D, it indeed varies the number of clients from 100 to 1000, but it only reports the test loss on MNIST. This is not convincing.

**Minor Comments**
1. Failed to disclose the usage of large language model as required.

---

> ### Author Response · Authors · 2025-11-27
>
> **W1: The experiments choose some relative simple datasets to validate its effectiveness. For example, for minist, cifar-10, and FMNIST, several baseline models already achieve test 100% accuracy. It's unclear how the methods perform on relative challenging datasets like Tiny-ImageNet.**
>
> Thank you for the time and effort you put into reviewing our paper and for your insightful comments and suggestions.
>
> Regarding the claim that baselines achieve 100% accuracy: This might be true in a centralized case using the plain datasets, but it is not true in the federated setting and under the various skews and mixings that we apply to the data. In particular, the tables in our paper, e.g. Table 1, show that the baselines achieve well below 100% accuracy on these datasets. For example, under concept shift, we see that IFCA achieves 52.7% accuracy for CIFAR-10 and 80.0% for MNIST. This is the case in the original IFCA paper as well: IFCA’s Table 1 shows that IFCA achieves only 81.51% accuracy on rotated CIFAR-10 with 2 rotations. This shows that these datasets, especially under the different types of skews are actually challenging in the federated setting, and this is why we focused on them. Regarding using Tiny-ImageNet, we believe though that CLoVE would still perform well with that dataset. However, note that none of the CFL baselines use Tiny-ImageNet and this is why we did not use it in our experiments either, as we would not be able to compare our ARI with that of the baselines.
>
> **W2: My concerns are on the communication costs and storage costs. Before the clustering stabilized, in each communication round, the server needs to broadcast K^(t) models and the clients need to store them. Compared to the FedAvg, the communication costs may be averaged out if the stabilization happens during the early stages and the algorithm convergence faster than FedAvg. Theorem 4.1 may justify for the simple scenario, yet CLoVE's actually performance in recovering the clusters is unclear. Maybe showing the training dynamic can help to justify. Nonetheless, the peak memory usage can still be large. Even though IFCA method, also suffers from this. I think this should be clearly discussed.**
>
> Your understanding is correct: the server can track the stability of client-to-cluster assignments, and once these assignments remain stable over several iterations, it can stop sending all $K$ models to each client and instead send only the model corresponding to each client’s cluster (and thereafter each cluster essentially performs regular FedAvg). In practice, as shown in our experiments, CLoVE achieves clustering stability with 100% accuracy within 2 rounds with high probability. This means that for a few rounds in the beginning there is some computation/communication burden, but this is what allows CLoVE to achieve so fast convergence and so high performance in recovering the clusters (ARI), as shown through our experiments in all Tables. Moreover, the training dynamic can be seen in Figure 6 of the revised paper (previously Figure 5), where again it can be seen that CLoVE converges to the correct clusters within 3-5 rounds, as well as from Tables 5, 10, 12 and 14, where one can see the ARI reached in 10 rounds and the first round when the ARI is more than 0.9, both showing CLoVE’s rapid improvement and fast convergence. The aforementioned Tables concern supervised experiments. To demonstrate CLoVE’s fast convergence even more clearly in the unsupervised setting as well, in the revised version we include the newly added Appendix D.4 (“Convergence Dynamics”) and Figure 7. As the figure show, CLoVE reaches a perfect ARI in just a few rounds in this setting as well.
>
> To address your concern about the peak memory usage, this can be alleviated as follows: the $K$ models do not need to be evaluated by the client all at the same time. Instead, if memory is a constraint, the client can be requesting and evaluating each of the $K$ sequentially one at a time, storing in memory only the loss produced by each model. This way, it can form the loss vector while only holding one model in memory at a time. We have added this remark in Appendix B.1 (“Communication and Computation Costs”) in the revised version and thank you for pointing this out.

---

> ### Author Response · Authors · 2025-11-27
>
> **Q1: In Algorithm 1, what are the conditions to determine if the clustering is stabilized?**
>
> Thank you for the question. The details for determining cluster stability were not included in the pseudocode for the sake of the algorithm’s readability and due to space constraints, but were included in Appendix D.1.5 (“Early Stopping of Clustering”). We understand that the paper would benefit if this was explained in the main part of the paper, so we added this explanation in the description of the pseudocode in Section 3 of the revised paper.
>
> Specifically, each client keeps the history of its assignment to models over the different rounds. Once a client determines that its historical assignments have been stable for a set number of rounds required for stability, the client declares to the server that it has reached stability. This happens at every round. The server at each round checks whether all (or a desired percentage) of the participating clients have reached stability, and if so, it decides that universal stability has been achieved. Note that in this way stability can be determined without the need for the server to maintain any client-specific state, thus conforming with the standard practice in federated learning.
>
> **Q2: Is CLoVE scalable to large number of clients? The main paper mentioned that "We vary the number of clients between 20 and 1000." However, in appendix C, the number of clients ranging from 12 to 50. In the appendix D, it indeed varies the number of clients from 100 to 1000, but it only reports the test loss on MNIST. This is not convincing.**
>
> Thank you for the question. As you mention, we had included scaling experiments in the unsupervised case in (former) Appendix D.2.2, with the number of clients varying from 100 to 1000. We complement these results with several new supervised experiments on three datasets: in the revised version we moved the unsupervised results to a separate Appendix D.3 (“Scaling with the number of clients”), and added Table 15 comparing the final ARI and the accuracy achieved by CLoVE and IFCA on MNIST, CIFAR-10 and FMNIST for the number of clients varying from 100 to 1000 in the supervised setting. We observe that CLoVE has no problem in scaling to a large number of clients in the supervised case as well, for all three datasets. Additionally, we added Figure 7, which shows the convergence behavior of CLoVE’s ARI over the course of its execution. We observe that CLoVE achieves perfect ARI after only a few rounds for different numbers of clients.
>
>
> **Failed to disclose the usage of large language model as required.**
>
> Thank you for pointing this out. According to the ICLR Author guide at https://iclr.cc/Conferences/2026/AuthorGuide, “if LLMs played a significant role in research ideation and/or writing to the extent that they could be regarded as a contributor, then authors should describe the precise role of the LLM in a separate section on LLM usage”. Since this does not apply in our case, we interpreted this guideline as not applicable and hence we did not add a corresponding section. However, if you believe we should, we would be happy to do it.

---

> > ### Comment · Reviewer_8Ay6 · 2025-11-27
> > **Follow-up Response from Reviewer**
> >
> > I would like to thank the authors for their detailed response. My major concerns are addressed, and I am willing to maintain my positive score.

---

### Official Review · Reviewer_MEzt · 2025-10-31

**Soundness:** 2
**Presentation:** 2
**Contribution:** 2
**Rating:** 4
**Confidence:** 3

**Summary:**

CLoVE introduces a paradigm for personalized federated learning through clustering of loss vector embeddings. By analyzing patterns of client losses across different models, the method achieves accurate client partitioning and collaborative optimization of cluster-specific models. Extensive experiments demonstrate that CLoVE rapidly converges in both supervised and unsupervised tasks, outperforming existing methods in clustering accuracy and model performance.

**Strengths:**

1. Proposes an approach using loss vector embeddings for client clustering, eliminating need for careful initialization.
2. Provides rigorous convergence analysis for mixed linear regression, to Theoretical guarantees for cluster recovery
3. Works in both supervised and unsupervised settings, with Simplicity and wide applicability

**Weaknesses:**

1. Analysis restricted to convex setting (linear regression), lacking guarantees for non-convex settings  commonly used in practice.
Missing comparisons with recent (2024-2025) state-of-the-art methods addressing similar challenges .
2. For the key challenge of sparse client participation in FL, the paper lacks corresponding theoretical analysis and systematic experimental validation.  For example, will the different setting of initicial K cluster number result in different results.
3. Insufficient experimental validation of dynamic cluster number selection method and sensitivity to initial cluster count. The impact of different initial cluster numbers on the final performance has not been systematically studied through ablation experiments. This fails to assure that the algorithm is insensitive to the initial choice of K, which is a significant concern in practical applications.

**Questions:**

Please see weaknesses.

---

> ### Author Response · Authors · 2025-11-27
>
> **W1: Analysis restricted to convex setting (linear regression), lacking guarantees for non-convex settings commonly used in practice. Missing comparisons with recent (2024-2025) state-of-the-art methods addressing similar challenges.**
>
> Thank you for the time and effort you put into reviewing our paper and for your insightful comments and suggestions.
>
> Our primary objective is to provide a rigorous analytical proof that CLoVE rapidly recovers the true client clusters. In particular, Theorem 4.1 shows that, with high probability, the correct clustering is achieved after a single iteration of CLoVE. However, as you have observed, we currently lack an analogous guarantee for the non‑convex setting despite our efforts in this direction. This gap reflects a fundamental difficulty: rapid‑convergence results in non‑convex settings are scarce in the literature. Existing baselines such as IFCA achieve fast convergence only when the underlying optimization problem is convex, and recent work (e.g., the ICML 2025 “FedGWC” paper that reviewer 3G9C points to) provides recovery guarantees only in the asymptotic regime as the number of rounds grows without bound. Note that all of our empirical results (except for the ones in Appendix D.1.1 which is by design dedicated to the linear setting) are for the non-convex setting, which clearly shows that CLoVE is able to handle a variety of non-convex settings very successfully in practice. Providing formal guarantees in non-convex settings is a central focus of our planned future research.
>
> Regarding the choice of baselines, we picked the particular ones because they are the most cited and thus well-established in the literature. We additionally added in the revised paper (Appendix B.3) a discussion of the ICML 2025 “FedGWC” paper that reviewer 3G9C points to.
>
> **W2: For the key challenge of sparse client participation in FL, the paper lacks corresponding theoretical analysis and systematic experimental validation. For example, will the different setting of initicial K cluster number result in different results.**
>
> **W3: Insufficient experimental validation of dynamic cluster number selection method and sensitivity to initial cluster count. The impact of different initial cluster numbers on the final performance has not been systematically studied through ablation experiments. This fails to assure that the algorithm is insensitive to the initial choice of K, which is a significant concern in practical applications.**
>
> Thank you for these comments. We realize that our description could be improved, and we changed it slightly for better clarity. Specifically, at each round, our algorithm determines the number of clusters that the loss vectors will be clustered into, using the silhouette score. To do so, CLoVE only needs an upper bound $\hat{K}$ for the search space for values of $K$ for which the silhouette score is evaluated. Given the upper bound, CLoVE will pick the best $K$ for the particular dataset up to that upper bound. In order to better clarify this in the main part of the paper, in the revised version we have added pseudocode in Algorithm 2 to describe in detail the process of the selection of the number of clusters, and have also edited Algorithm 1’s description in Section 3 accordingly.

---

### Official Review · Reviewer_ymcK · 2025-11-01

**Soundness:** 2
**Presentation:** 3
**Contribution:** 2
**Rating:** 4
**Confidence:** 3

**Summary:**

The paper proposes a clustered PFL method that embeds each client by its vector of losses on multiple candidate models, then clusters these embeddings to assign clients and train per-cluster models. By iterating between “compute loss vectors → cluster clients → federated update per cluster,” it avoids IFCA-style sensitivity to initialization and does not require inter-cluster separation assumptions. The paper proves single-shot cluster recovery and exponential convergence in a linear regression setting and reports fast, accurate clustering across supervised and unsupervised non-IID benchmarks. Empirically, CLoVE converges in few rounds and tolerates partial participation/stragglers.

**Strengths:**

Using loss-vector embeddings sidesteps careful warm-starts and delivers quick, stable clustering in practice.

**Weaknesses:**

1. guarantees are shown only for linear models; applicability to nonconvex deep networks remains unproven.

2. clients must evaluate multiple models each round to form loss vectors, which increases local compute/communication and may leak information about client data through loss profiles.

3. dynamic clustering with sparse participation, label noise, or malicious clients may oscillate or be exploitable, and robustness is not theoretically characterized.

**Questions:**

1. What formal guarantees (if any) extend to nonconvex settings—e.g., bounds under smoothness/PL conditions—or at least empirical ablations isolating architecture depth/width?

2. How does per-round cost scale with the number of clusters KKK, and can you reduce evaluations (e.g., sub-sampling models, proxy losses, or distillation) while preserving clustering accuracy?

3. What privacy defenses (secure aggregation, DP on losses, or randomized response) are compatible with CLoVE, and how do they trade off against clustering fidelity and convergence under client drift or adversarial behavior?

---

> ### Author Response · Authors · 2025-11-27
>
> **W1: guarantees are shown only for linear models; applicability to nonconvex deep networks remains unproven.**
>
> **Q1: What formal guarantees (if any) extend to nonconvex settings—e.g., bounds under smoothness/PL conditions—or at least empirical ablations isolating architecture depth/width?**
>
> Thank you for the time and effort you put into reviewing our paper and for your insightful comments and suggestions.
>
> Our primary objective is to provide a rigorous analytical proof that CLoVE rapidly recovers the true client clusters. In particular, Theorem 4.1 shows that, with high probability, the correct clustering is achieved after a single iteration of CLoVE. However, as you have observed, we currently lack an analogous guarantee for the non‑convex setting despite our efforts in this direction. This gap reflects a fundamental difficulty: rapid‑convergence results in non‑convex settings are scarce in the literature. Existing baselines such as IFCA achieve fast convergence only when the underlying optimization problem is convex, and recent work (e.g., the ICML 2025 “FedGWC” paper that reviewer 3G9C points to, a discussion of which we added in Appendix B.3) provides recovery guarantees only in the asymptotic regime as the number of rounds grows without bound. Note that all of our empirical results (except for the ones in Appendix D.1.1 which is by design dedicated to the linear setting) are for the non-convex setting, which clearly shows that CLoVE is able to handle a variety of non-convex settings very successfully in practice. Providing formal guarantees in non-convex settings is a central focus of our planned future research.
>
> **W2: clients must evaluate multiple models each round to form loss vectors, which increases local compute/communication and may leak information about client data through loss profiles.**
>
> **Q3: What privacy defenses (secure aggregation, DP on losses, or randomized response) are compatible with CLoVE, and how do they trade off against clustering fidelity and convergence under client drift or adversarial behavior?**
>
> Thank you for this observation and the question. A detailed discussion of CLoVE’s computation and communication cost is provided in Appendix B.1 (“Communication and Computation Costs”). CLoVE’s early stopping feature, all models need only to be sent to all clients for the rounds until stability is reached. As shown in our experiments (see App. D.1.5), stability is achieved in a few rounds with high probability. After that, only a single model needs to be sent to each client. From then on, each cluster performs regular FedAvg, so the cost is similar to FedAvg, which is the best possible (other baselines also have a computation and communication cost at least as high as that of FedAvg). Additional cost savings can be achieved by leveraging weight-sharing techniques from multi-task learning, as mentioned in Appendix B.1.
>
> Additionally, compared with other PFL methods like IFCA that exchange only model weights, CLoVE additionally sends the loss vectors. The size of the loss vectors is $\Theta(K)$ per client. This is small compared to the model weights themselves, which are $\Theta(K*R)$ per client, where $R$ is the number of model parameters (or gradients). Also, in practice, $K$ is usually small. So the overhead of communicating the loss vectors is not significant.
>
> Regarding a potential privacy leakage through loss profiles: As noted in the privacy discussion included in Appendix B.2, privacy is not the focus of this paper. That said, the model loss sent by each client is aggregated information, since it is an average over the entire client’s dataset. While this aggregate measure is generally not considered privacy-sensitive, its privacy implications remain underexplored. Further research is needed to assess whether, and under what conditions, such aggregate statistics could lead to unintended information leakage. Moreover, several baselines (e.g. Tan et al. (2022); Xu et al. (2023); Vahidian et al. (2023)) similarly share 1D vectors or summaries of client data (we only send losses, not summaries of data). This process is deemed irreversible by those baselines, and they even mention that further privacy-preserving techniques can be employed.

---

> ### Author Response · Authors · 2025-11-27
>
> **W3: dynamic clustering with sparse participation, label noise, or malicious clients may oscillate or be exploitable, and robustness is not theoretically characterized.**
>
> Regarding dynamic clustering with sparse participation: this is a setting that CLoVE is able to handle successfully. This is shown in the experiments of Appendix D.1.6 (“Partial Participation”), where CLoVE’s accuracy and ARI are both near optimal regardless of the level of participation, which is varied from 100% to as low as 10%. Regarding label noise: although we did not study the particular case of noisy labels and we appreciate the suggestion for our future work, we did study many types of label skews, feature skews and concept shifts that show CLoVE’s robustness to challenging settings. The description of the settings is provided in Section 5.1 “Data Partitioning” and Appendix D.1.2 (“Additional Label Skews”). The study of malicious behavior and theoretical robustness with respect to these suggestions cannot fit in this paper given its already extended length, especially given that the majority of the baselines do not specifically include them either, but these are very interesting directions for future work and we appreciate the suggestions.
>
> **Q2: How does per-round cost scale with the number of clusters KKK, and can you reduce evaluations (e.g., sub-sampling models, proxy losses, or distillation) while preserving clustering accuracy?**
>
> Until convergence happens, the per-client computation cost is directly proportional to the number of clusters, as the clients evaluate one model per cluster. After convergence, which usually happens in only a few rounds, the per-client cost is the same as FedAvg: the client only trains a single model, the one that has been assigned to its cluster. A more thorough discussion of the cost behavior can be found in Appendices B.1 (“Communication and Computation Costs”) and D.1.5 (“Early Stopping of Clustering”).
>
> Regarding your other suggestions: we do already support and have evaluated subsampling of clients (see Alg. 1 and Appendix D.1.6 (“Partial Participation”)). However, we realize the readability of the algorithm would improve if we stated explicitly that the partial participation percentage is a parameter that the server can choose, as this is the exact parameter that we are varying in the experiments of Appendix D.1.6 anyway. We have modified our pseudocode to reflect that parameter. Note that our results of Table 14 and the newly added experiments of the new Figure 4 show that CLoVE still performs very well even for very low participation rates (e.g. 10%).

---

### Author Response · Authors · 2025-11-27

We would like to share with the reviewers and the ACs our anonymized code repository, timestamped before the submission deadline: https://anonymous.4open.science/r/CLoVE-PFL. We plan to release the code publicly to aid other researchers after acceptance.

---

### Meta-Review · Area_Chair_DiEU · 2026-01-07

**Summary:**

The paper proposes CLoVE, a clustered federated learning (CFL) method that clusters clients based on loss vector embeddings derived from evaluating multiple candidate models on local data. The method iteratively refines clusters and updates cluster-specific models, aiming to avoid sensitivity to initialization and support both supervised and unsupervised settings. Reviewers generally acknowledge the algorithm’s simplicity, empirical effectiveness in fast cluster recovery, and applicability across non-IID settings. However, consistent concerns were raised regarding: (1) the lack of theoretical guarantees beyond linear (convex) settings; (2) limited validation on more challenging or large-scale datasets; (3) insufficient ablation on sensitivity to the initial number of clusters K; and (4) communication/computation overhead during the clustering phase. One reviewer (3G9C) expressed stronger skepticism about contribution and baseline comparisons, while another (8Ay6) initially raised scalability and cost concerns but was largely satisfied by the rebuttal.

**Reviewer Concerns:**

Addressed concerns:

- Reviewer 8Ay6: Concerns about communication/storage costs and scalability were adequately addressed. Authors clarified early stopping behavior, provided new scaling experiments (Table 15, Figure 7), and explained memory-efficient evaluation strategies. The reviewer explicitly stated their major concerns were resolved.
- Reviewer MEzt & ymcK: Questions about sparse participation and dynamic K were partially addressed. Authors added Algorithm 2 and clarified that K is selected via silhouette score within a bounded range, and showed robustness to low participation (Appendix D.1.6). However, systematic ablation on sensitivity to the upper bound of K remains limited.
- Reviewer 3G9C: The comparison with FedGWC (a 2025 CFL method) was added in Appendix B.3, and clarification on Dirichlet-based partitioning was incorporated into the main text. The degenerate cluster concern was reasonably mitigated by noting that single-client clusters may be optimal and do not inherently leak private data.


Outstanding concerns:

- Theoretical limitations: All reviewers noted the absence of non-convex convergence guarantees. While authors correctly point out this is a common gap in the literature, the lack of empirical ablation isolating architecture depth/width remains unaddressed.
- Baseline completeness: Despite adding FedGWC discussion, the experimental comparison still omits several recent (2024–2025) PFL/CFL methods, as flagged by MEzt.
- Accuracy vs. ARI trade-off: Reviewer 3G9C’s observation that CLoVE sometimes underperforms in accuracy despite high ARI is acknowledged by authors but not rigorously analyzed—this tension is noted but not resolved.

**Reviewer Scores:**

Reviewer ymcK: Original score: 4. The rebuttal clarifies computational cost and privacy considerations, and acknowledges theoretical gaps honestly. Given the responsiveness and partial mitigation of concerns (e.g., cost analysis), the reviewer would likely maintain or slightly increase their score. Estimated post-rebuttal score: 4 or 6.

Reviewer MEzt: Original score: 4. The addition of Algorithm 2 and clarification on K selection addresses the vagueness concern. However, the lack of ablation on K sensitivity and missing recent baselines remain. Estimated post-rebuttal score: 4 (no change).

Reviewer 8Ay6: Original score: 6. Explicitly stated concerns were addressed, and new results strengthen scalability claims. Estimated post-rebuttal score: 6 (no change).

Reviewer 3G9C: Original score: 2. While the rebuttal adds useful context (FedGWC comparison, Dirichlet clarification), core concerns about contribution novelty, accuracy performance, and theoretical scope persist. No indication of score revision. Estimated post-rebuttal score: 2 or 4.

---

### Decision · Program_Chairs · 2026-01-26

Reject